# Interaction-driven transport of dark excitons in 2D semiconductors with phonon-mediated optical readout

Saroj B. Chand[1], John M. Woods[1], Jiamin Quan[1], Enrique Mejia[1], Takashi Taniguchi [2], Kenji Watanabe [3], Andrea Alù [1,4,5] & Gabriele Grosso [1,5] ✉

The growing field of quantum information technology requires propagation of information over long distances with efficient readout mechanisms. Excitonic quantum fluids have emerged as a powerful platform for this task due to their straightforward electro-optical conversion. In two-dimensional transition metal dichalcogenides, the coupling between spin and valley provides exciting opportunities for harnessing, manipulating, and storing bits of information. However, the large inhomogeneity of single layers cannot be overcome by the properties of bright excitons, hindering spin-valley transport. Nonetheless, the rich band structure supports dark excitonic states with strong binding energy and longer lifetime, ideally suited for long-range transport. Here we show that dark excitons can diffuse over several micrometers and prove that this repulsion-driven propagation is robust across non-uniform samples. The long-range propagation of dark states with an optical readout mediated by chiral phonons provides a new concept of excitonic devices for applications in both classical and quantum information technology.

Since the discovery of excitonic resonances in $MoS_2$[1,2], transition metal dichalcogenides (TMDs) have emerged as a promising platform for applications in classic and quantum technology. The atomic-size thickness of TMDs imbues excitons with a unique combination of properties that make them attractive in many scientific fields, including optoelectronics, sensing, quantum information, and spintronics[3]. The large binding energy makes excitons in TMDs robust at room temperature, and the combination of their short lifetime and in-plane dipole makes them extremely bright. The broken inversion symmetry and large spin–orbit coupling gives rise to a unique spin-valley coupling that provides an extra degree of freedom for manipulating and storing information[4]. These aspects have been mostly explored in bright excitons, which are bound states formed by an electron and a hole in electron bands with the same spin orientation at the K (or K')

valley. On one hand, bright excitons are optically active, with a large oscillation strength and high quantum yield. However, their short lifetime, weak interaction, and short transport range are severe limitations for the potential of TMDs in spintronics and quantum technology[5]. On the other hand, the band structure of TMDs gives rise to a variety of excitonic complexes[6], and it allows for the formation of several lower-energy dark excitons that cannot directly recombine optically due to the non-conservation of spin or momentum. Spin-forbidden excitons are made of carriers in electron bands with opposite spin orientations in the same valley (K and K'), and contrary to their bright counterpart, they possess an out-of-plane transition dipole and two orders of magnitude longer lifetime[7–11]. Despite their dark nature, spin-forbidden excitons can inefficiently recombine optically due to the small spin crossover of the conduction band or through the Rashba

[1]Photonics Initiative, Advanced Science Research Center, City University of New York, New York, NY 10031, USA. [2]International Center for Materials Nanoarchitectonics, National Institute for Materials Science, 1-1 Namiki, Tsukuba 305-0044, Japan. [3]Research Center for Functional Materials, National Institute for Materials Science, 1-1 Namiki, Tsukuba 305-0044, Japan. [4]Department of Electrical Engineering, City College of the City University of New York, New York, NY 10031, USA. [5]Physics Program, Graduate Center, City University of New York, New York, NY 10016, USA. ✉e-mail: ggrosso@gc.cuny.edu

effect[12]. Moreover, the spin-flip process required for dark exciton recombination can be controlled by an external magnetic field[13,14]. Although the dynamics and transport of interlayer excitons in van der Waals heterostructures[15–17] and intervalley excitons[18] have been recently investigated, the dynamics and the transport properties of spin-forbidden dark excitons have not been fully explored yet[19], due to the challenges associated with their detection using far-field spectroscopy techniques[11,20,21]. However, efficient phonon interaction processes can open recombination paths for dark excitons with out-of-plane emission that allow for efficient optical readout without the need for auxiliary external fields.

In this work, we study exciton dynamics by employing high-resolution, spatially-resolved photoluminescence (PL) spectroscopy techniques that enable the observation of their transport up to several micrometers. We demonstrate the long-range transport of dark excitons by resolving spectrally their emission away from the excitation position. We show that dark exciton transport is driven by mutual repulsive interaction, and it depends on the exciton density at the excitation region (see Fig. 1a). The information of the quantum state of dark excitons is then read out via the emission of lower-energy excitons resulting from exciton-phonon scattering. Our conclusions are corroborated by two main observations: (i) the size of the dark exciton cloud increases as a function of density; and (ii) dark excitons are shown to diffuse in different energy landscapes, including flat, downhill, and uphill. This last observation can be explained by the strong repulsive interaction energy that overcomes the energy difference between the excitation and detection point. The larger interaction

energy of dark excitons compared to the bright counterpart is explained by their higher density. The cartoon in Fig. 1a illustrates the main mechanisms and findings of our experiments. We note that this robust transport is peculiar to dark excitons as, differently from interlayer excitons in heterostructures, the stronger binding energy combined with high interaction energy can compensate for largely inhomogeneous energy landscapes. Moreover, this effect is clearly observable in naturally n-doped samples, indicating that it can be used in simple device geometries without the need for external gating or complex heterostructures in which the exciton properties strongly depend on the twist angle.

## Results

### PL spectroscopy of encapsulated WS₂ monolayer

Figure 1b (bottom panel) shows the measured emission spectrum at $T = 7$ K of the WS₂ monolayer encapsulated within two thin hBN layers. In order to study the spin-valley coherence properties and identify the several excitonic complexes in the sample[22,23], we excite it with a right-handed circularly polarized ($\sigma^+$) laser quasi-resonant with the $B$ exciton and detect the emission of right-handed circularly polarized ($\sigma^+$) or left-handed circularly polarized ($\sigma^-$) light. The top panel shows the chirality, defined as the degree of polarization and calculated as $\rho = \frac{I^+ - I^-}{I^+ + I^-}$, where $I^+$ and $I^-$ is the emission intensity of $\sigma^+$ and $\sigma^-$ polarized light, respectively. The sample shows several excitonic peaks whose energy and polarization degrees are in good agreement with previous studies in n-doped WSe₂ and WS₂[22–24]. Peaks labeled with $X^0$, $XX^0$, $X_T^-$, $X_S^-$, and $XX^-$ are the bright $A$ neutral exciton, neutral

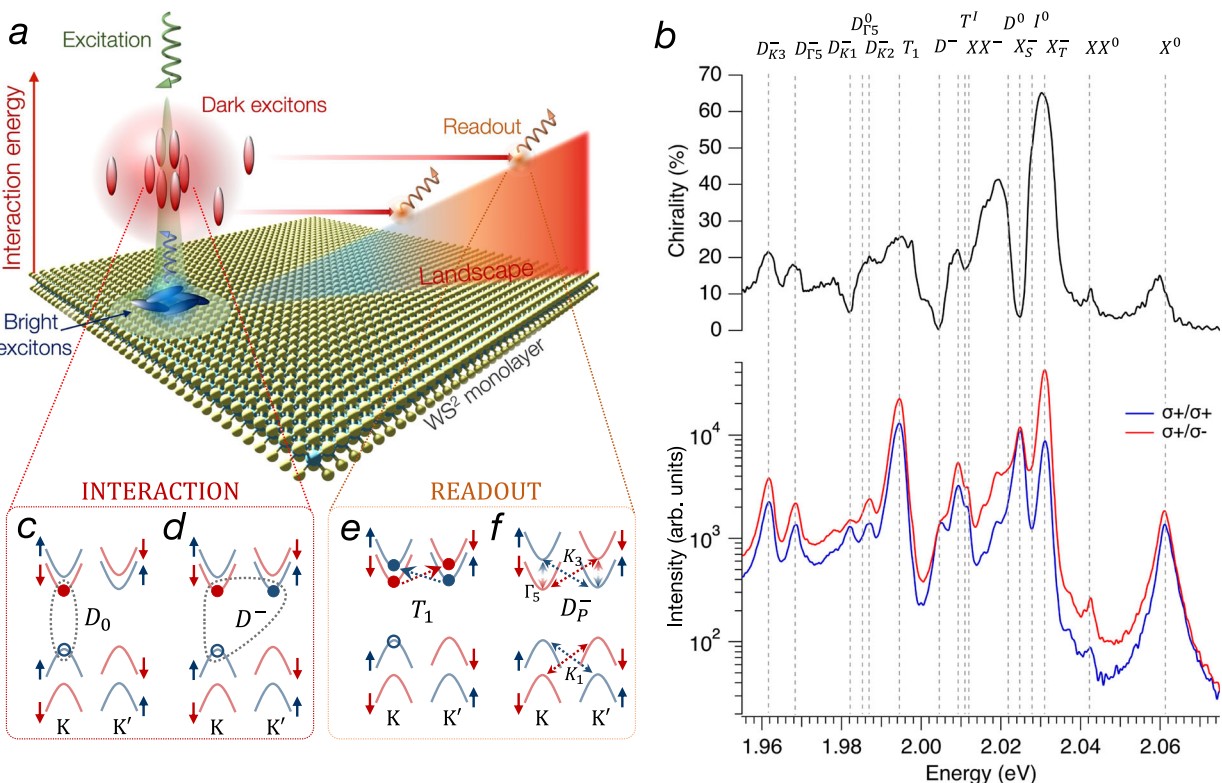

**Fig. 1 | Bright and dark exciton complexes in monolayer WS₂. a** Upon irradiation, exciton density follows a Gaussian distribution imprinted by the laser. In the high-density region, the energy of dark excitons increases due to the repulsive interaction. The extra interaction energy drifts dark excitons allowing them to propagate over large areas of the sample and optically recombine far away from the excitation spot. The rainbow ramp represents the energy landscape used in our experiments in which the energy increases linearly due to strain. The energy of bright excitons does not change significantly due to the smaller density and, as a

result, they do not drift and their emission is limited to the region of the laser excitation. **b** Emission spectra of WS₂ at $T = 7$ K excited with $\sigma^+$ circular polarization and collected with $\sigma^+$ (blue line) and $\sigma^-$ (red line). The top panel shows the chirality of the emission $\rho = \frac{I^+ - I^-}{I^+ + I^-}$, where $I^+$ and $I^-$ are the emission intensity of $\sigma^+$ and $\sigma^-$ polarized light, respectively. The peaks of bright and dark exciton complexes are highlighted by vertical dashed lines. The band compositions of the relevant dark exciton species are illustrated in the cartoon in (**c**–**f**).

biexciton, triplet trion, singlet trion, and negatively charged biexciton. These exciton complexes are the results of the Coulomb interaction between $X^0$ and carriers in the K or K' valley. The energy detuning with respect to the bright exciton of all exciton complexes measured in this work is in excellent agreement with previous works and is reported in Supplementary Tables S1 and S3. We identify two peaks in the low-energy shoulders of the bright trions that have been previously attributed to intervalley momentum-forbidden ($I^0$) and intravalley spin-forbidden ($D^0$) excitons[23,25]. The energy difference between $I^0$ and $D^0$ is the result of short-range Coulomb exchange interactions in $WS_2$, measured to be ~ 6.5 meV in agreement with theoretical predictions[26]. The presence of dark excitons is more evident in the chirality plot (top panel of Fig. 1b), where clear features emerge at the low energy side of $X_T^-$ and $X_S^-$. We note that the chirality of $D^0$ should be suppressed due to its out-of-plane optical transition[23]. The nonvanishing chirality of $D^0$ that emerges in our measurements is due to the nearby presence of the strong emission of $X_S^-$ and of a strongly-polarized unknown peak at ~ 3 meV below. A more careful analysis returns a much lower degree of polarization for $D^0$ of 7% (see Section 3 of the SI). The peaks labeled with $T^I$ and $D^-$ have been attributed to the intervalley momentum-forbidden and intravalley spin-forbidden dark trions[22-24]. The peak $T_1$ was recently ascribed to the optical recombination of $D^-$ mediated by electron-electron intervalley scattering[22,27]. Several peaks are observed in the lower energy side of the spectrum and are attributed to the replicas of $D^-$ and $D^0$ due to the interaction with chiral phonons. We identify $D_i^-$ as replicas of $D^-$ generated by the interaction with valley phonons $i = K_1, K_2, K_3, \Gamma_5$[23,25,28,29]. Theoretical calculations of the energy of these phonon modes, the rationale for the assignment of these peaks, and the discussion on additional peaks observed in the experimental spectrum are reported in the Supplementary Information. The band composition of the dark exciton complexes relevant for the rest of the discussion is illustrated in Fig. 1c–f. Note that the dynamics of $D_i^-$ and $T_1$ can be ascribed to the one of $D^-$ and $D^0$, but, differently from $D^-$ and $D^0$, their emission is out-of-plane[23] and they appear more clearly in the emission spectrum of Fig. 1b. In the text, we refer to $D^0$ and $D^-$ as dark excitons and dark trions if not otherwise specified.

## Diffusion of exciton complexes in WS₂

First, we study how different exciton populations diffuse by measuring the size of their cloud, obtained by spectrally filtering the corresponding peaks as discussed in the Supplementary Information. Figure 2a shows the map of the PL emission from the $WS_2$ monolayer used for this experiment taken by scanning the sample with galvanometer mirrors in confocal mode (see Fig. S1). During the sample fabrication process, we introduce tensile strain gradients in a large region of the monolayer to create an energy landscape for excitons. The details of the fabrication process and the characterization of strain are discussed later in the text and in the SI. The position in the sample and the direction of the energy gradient are indicated in Fig. 2a. Figure 2b–d is the spatially resolved emission of $X^0$, $T_1$ and $D_P^- = D_{K3}^- + D_{\Gamma5}^-$, respectively, taken with an EMCCD. Figure 2b shows that the extension of the bright exciton cloud vanishes completely around 4 μm away from the excitation location. The diffusion of bright excitons is driven by their non-uniform concentration at the excitation spot and it is constrained by the short lifetime, in the order of 1–5 ps, and weak interaction[30,31]. Differently from the bright exciton, the clouds of dark species extend over a larger area, and the emission is still detected more than 8 μm away from the excitation location, double the distance of its bright counterpart. A comparison of the intensity profiles is illustrated in Fig. 2e. Two main factors contribute to the longer diffusion of dark excitons: the long lifetime allows them to travel further distances, and the larger interaction energy generates a drift potential that pushes dark excitons away from the excitation location. These features are well suited for the requirements of long transport in quantum technologies.

## Density-dependent propagation of dark excitons in WS₂

The strong interaction of dark excitons is further demonstrated by power-dependent measurements. The evolution of the emission spectrum for increasing pump power, reported in Fig. S7 in the SI, shows a smaller redshift due to band renormalization for dark exciton species when compared to the bright counterparts[32-34]. This suggests that dark excitons are affected by stronger repulsive interactions. The effect of this interaction is manifested more clearly by the expansion of

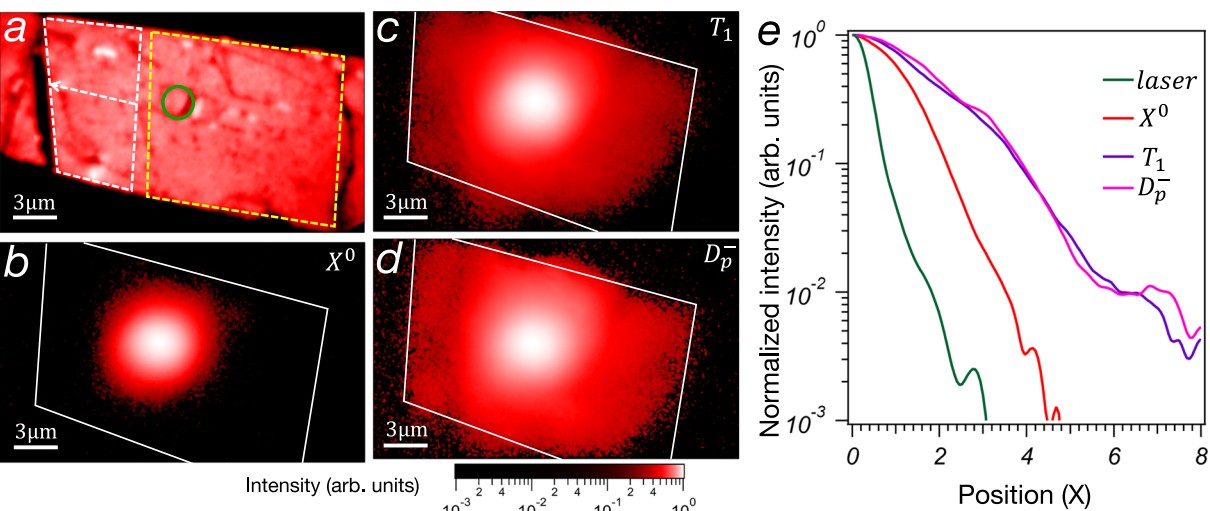

**Fig. 2 | Spatial emission of bright and dark excitons. a** Photoluminescence map of the $WS_2$ monolayer encapsulated within thin layers of hBN. White dashed lines indicate regions of the sample characterized by a linear strain gradient that creates increasing exciton energy in the direction of the arrow. Yellow lines indicate a sample region with no significant strain gradient. The green circle highlights the position of the laser for the diffusion measurement in (**b–d**). Images of the cloud of **b** bright excitons, **c** dark excitons $T_1$, and **d** dark trion phonon replica $D_P^- = D_{K3}^- + D_{\Gamma5}^-$ shows the diverse propagation dynamics. The edges of the $WS_2$ flake are highlighted by white lines. The images in **c–e** are acquired with a CW laser with power 600 μW. **e** Spatial intensity profiles along the positive horizontal direction for the laser, bright, and dark excitons.

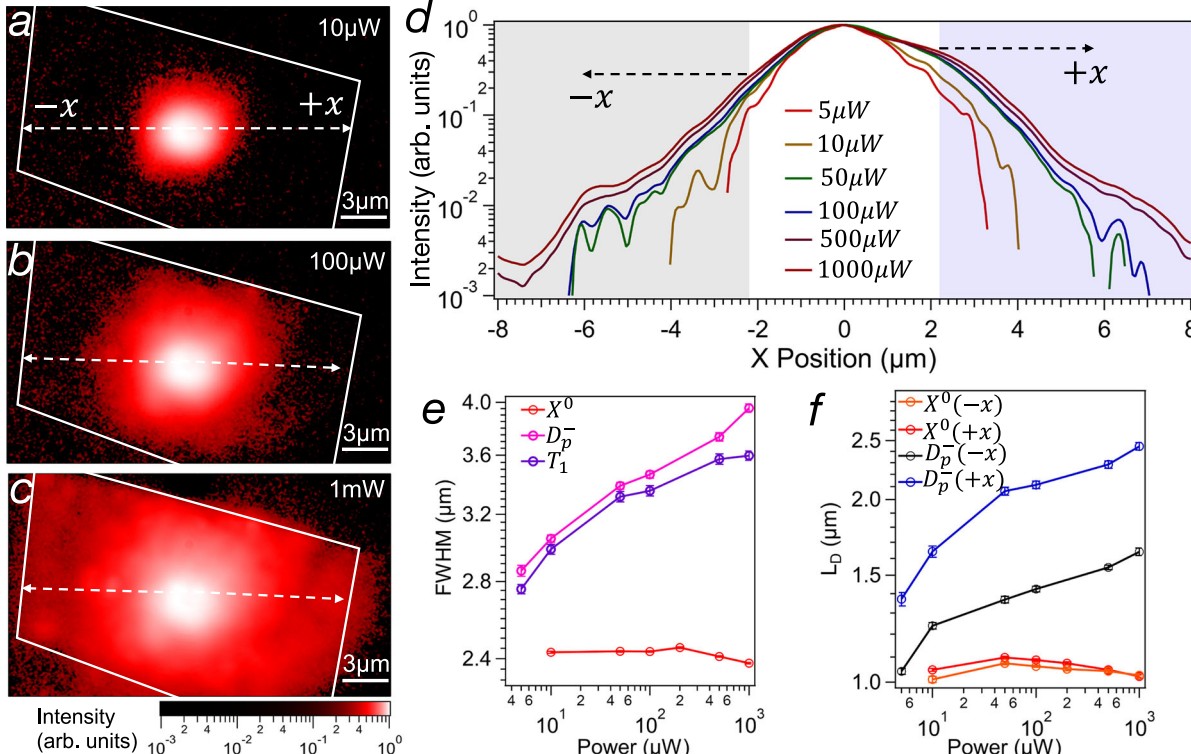

**Fig. 3 | Density-dependent propagation of dark excitons in WS₂. a–c** Power-dependent diffusion of the dark trion phonon replicas ($D_P^- = D_{K3}^- + D_{\Gamma5}^-$). **d** Intensity profiles at different excitation powers show as the cloud of $D_P^-$ expands due to increasing interaction. The colored areas indicate the regions used to extract the diffusion length along the strained ($-x$) and unstrained ($+x$) directions. **e** Comparison of the full-width half maximum (FWHM) of the exciton cloud of $X^0$, $D_P^-$, and $T_1$ as a function of the pump power. **f** Diffusion length of the of $X^0$ and $D_P^-$ as a function of the pump power. The error bars in **e** and **f** indicate the error of the fitting.

the dark exciton cloud. Figure 3a–c illustrates how the diffusion of $D_P^-$ increases when the pump power is increased by two orders of magnitude. The horizontal line profiles of the intensity plotted in Fig. 3d confirm the expansion of the dark exciton cloud with increasing exciton density corroborating the repulsive nature of exciton–exciton interactions. Exciton diffusion is evaluated by measuring the full-width half maximum (FWHM) of the spatial emission. Differently from $X^0$ in which the FWHM shows a constant value of 2.4 μm, the FWHM of dark excitons increases from 2.8 to 3.9 μm when the pump power is increased by two orders of magnitude. Diffusion measurements of $X^0$ and $T_1$, the filtering, and the fitting procedures are shown in Supplementary Figs. S8–S11. In addition, we estimate the characteristic diffusion length of dark excitons by fitting the line profiles away from the center of laser excitation with the two-dimensional diffusion model with a point source according to $n(r) \sim \frac{e^{-r/L_D}}{\sqrt{r/L_D}}$ [16,19,35]. To investigate the effect of exciton–exciton interaction, we monitor the diffusion length at different excitation powers. Figure 3f shows that $L_D$ increases from 1.4 to 2.4 μm in the unstrained ($+x$) directions and from 1 to 1.6 μm in the strained ($-x$) region. The asymmetry in the diffusion length in the strained uphill and unstrained directions is attributed to the different energy landscapes. Figure S12 in the SI shows the characterization of strain along the horizontal direction $x$. After a small interface region, in the negative $x$ direction, strain creates a smooth and linear uphill energy gradient that limits diffusion. In both directions, the increase of the characteristic diffusion length $L_D$ with increasing excitation power indicates an enhanced diffusion at high exciton densities. Similarly to interlayer excitons[16,17], exciton interactions promote longer diffusion. We note how the diffusion length of dark excitons is comparable to the one of interlayer excitons despite the shorter lifetime ($\tau$), suggesting a different interaction strength. As discussed below, we attribute the

increase in diffusion length of dark excitons to the large interaction energy. The diffusion constant ($D = L_D^2/\tau$) of dark excitons in the unstrained region ranges from 40 to 240 cm²/s, in agreement with previous reports[19]. Differently from dark excitons, the diffusion length of $X^0$ is not density-dependent and stays constant at around 1 μm for all pump powers. A similar value for the diffusion length of bright excitons has been already reported[19] and attributed to the combined contribution of a rapid and slower decay rate[36]. However, we note that, while our imaging method is capable of detecting very weak signals, point sources in the sample can appear diffuse, affecting the spatial resolution. A comparison with confocal imaging is discussed in Section 5 and Figure S11 of Supplementary Information.

### Exciton diffusion in different energy landscapes

The differential between the diffusibility of dark excitons and other excitonic complexes allows us to analyze how exciton diffusion interacts with a varying potential energy landscape. We note that, despite the large inhomogeneity of the sample (Fig. 2a and Fig. S12), long diffusion occurs in the dark exciton species, indicating robustness against energy barriers due to strain and impurities. To further prove that dark excitons are a resilient way to transport energy and information across samples, we study exciton diffusion in a linear energy gradient. Tensile strain is carefully produced in the encapsulated WS₂ monolayer during the transfer process (more details in the SI) such that a linear energy gradient is created in a region extending over 6 μm. Tensile strain reduces the energy band-gap ($E_{BG}$) at the K valley and transforms the energy landscape for excitons whose energy reads $E_X(x) = E_{BG}(x) - E_B$. Here we assume that the exciton binding energy ($E_B$) does not depend on the spatial coordinate $x$, because strain only weakly affects it in the strain range investigated in this work[37]. Uniaxial

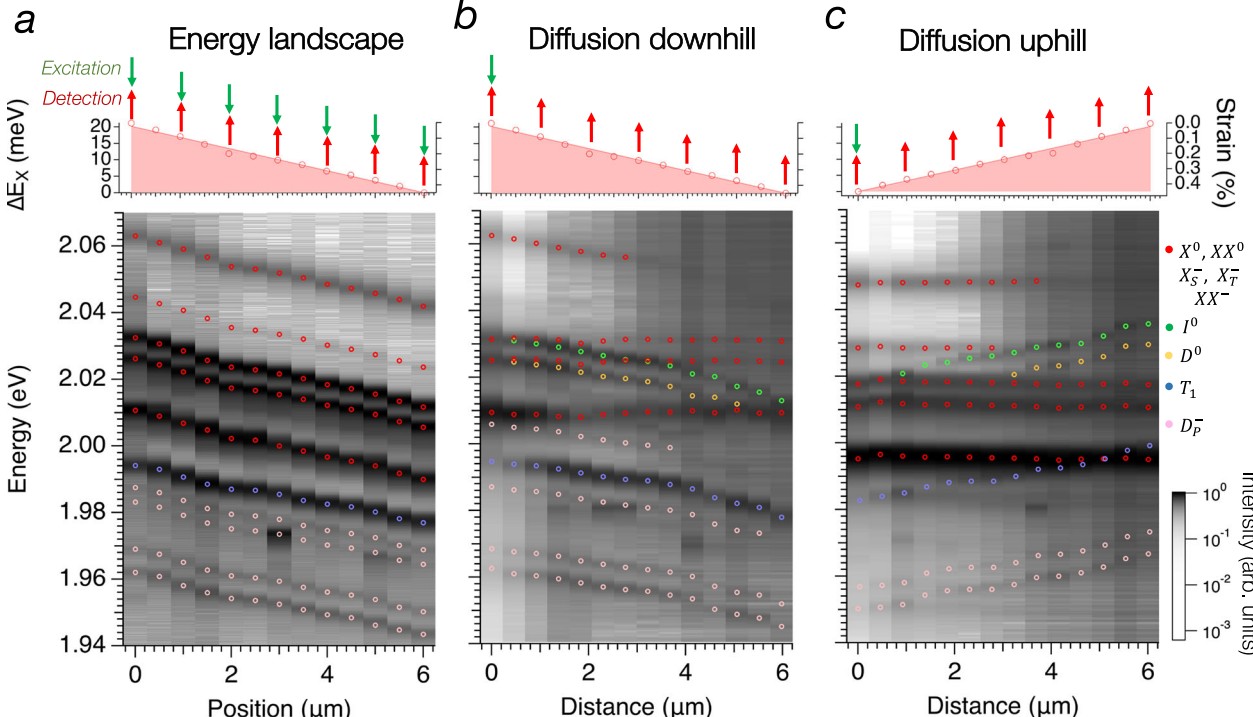

**Fig. 4 | Exciton diffusion in different energy landscapes. a** The energy landscape generated by strain is characterized by measuring the emission spectrum of the monolayer WS$_2$ in different positions. Excitation and detection are done at the same location as described in the top panel. The color map in the bottom panel shows the spectral emission normalized to the maximum at different positions. The energy corresponding to the exciton complexes is highlighted by circular markers of different colors: red ($X^0$, $XX^0$, $X_T^-$, $X_S^-$, $XX^-$), green ($I^0$), yellow ($D^0$), blue ($T_1$), and rose ($D_{K3}^-$ and $D_{T5}^-$). All exciton species show an energy shift up to 20 meV corresponding to a linear tensile strain gradient from 0 to 0.4%. **b** Downhill diffusion of

excitons created at the top of the potential gradient (top panel) and measured at increasing distances from the generation location. The diffusion of the dark excitons appears clearly from the spectral response as a function of the distance from the excitation (bottom panel). **c** Uphill diffusion of excitons created at the bottom of the potential gradient (top panel) and measured at increasing distances from the generation location. Differently from the downhill case, only dark excitons propagate far away from the excitation spot. After propagation due to repulsive interactions, dark excitons relax towards the bottom of their dispersion and recombine radiatively via different paths, generating $I^0$, $T_1$, and $D_P^-$.

strain variations in TMDs have been shown to modify the emission energy of the neutral excitons $X^0$ with a uniaxial strain gauge factor of −45 meV/%[38,39]. Although this value has been measured at room temperature in previous experiments, it is a good approximation to estimate the strain magnitude in our samples. Figure 4a shows the position-dependent PL spectra measured by exciting and collecting the emission spectra in the same position along one direction. Note that, for the sake of visibility, in all color maps of Fig. 4 the spectra (plotted as columns of the map) are normalized to their maximum, and the energy of bright (dark) exciton complexes is highlighted by red (other colors) circular markers. The variation of intensity of the peaks as a function of space can be appreciated in the raw spectra reported in Supplementary Fig. S13. Figure 4a shows that as the position moves in the direction of increasing applied strain, a clear linear energy shift up to 20 meV is visible for all excitons, confirming the presence of a one-dimensional linear strain gradient in the sample from 0 to 0.4%. This strain distribution allows us to study exciton diffusion in both downhill and uphill energy landscapes. We perform energy-resolved diffusion experiments by decoupling detection and excitation so that excitons are excited in one location and their emission spectrum is measured far away from it (see Supplemental Fig. S1). The schematic of the experiment is illustrated in the top panel of Fig. 4b, c. When excitons diffuse away from the excitation spot, they relax toward their minimum energy state before recombining radiatively[28] and their emission energy depends on the local conditions, including the variation of $E_{BG}$ due to strain (Fig. 1a). This process is illustrated in Fig. 5c.

First, we study downhill diffusion by creating the exciton gas in a point of the sample with zero strain corresponding to high potential energy and by measuring the emission spectra at different positions

along the strain gradient. Differently from other exciton species, the energy of the negatively charged biexciton ($XX^-$) and the bright trions ($X_T^-$ and $X_S^-$) does not change over the entire strain landscape. Due to their low binding energy ($\sim 20-40$ meV), very short lifetime ($\sim 30$ ps), and large mass[24,40], the diffusion of these species is limited. However, at the excitation location, they have the strongest emission (see Fig. S7), which can be picked by our large NA objective lens even at a distance of a few micrometers. This conclusion is supported by the constant emission energy, showing that the recombination occurs only in the same location of the excitation. However, we observe significant diffusion of bright excitons up to 2.5 μm, in agreement with the diffusion measurements of Fig. 2c. Despite its short lifetime, $X^0$ has large binding energy, and the tensile strain generates a funneling effect similar to what was observed in previous works[41]. Very good correlation emerges between the energy landscape created by the strain gradient and the emission energy of dark exciton complexes, including $I^0$, $D^0$, $D^-$, $T_1$, $D_P^-$. The downhill transport of dark excitons is due to the combination of repulsion interaction and the funneling force $\mathbf{F}_{fun} = -\boldsymbol{\nabla} E(x)$ generated by the strain gradient.

Finally, we study uphill diffusion by exciting the sample in a location with high strain, corresponding to low potential energy. Figure 4c shows that the emission energy of all bright excitons detected away from the excitation spot is constant, despite the increase of potential energy. This indicates that they are not propagating, and they are observed in the spectrum only because the large NA objective lens picks up their strong out-of-plane emission at the excitation spot. This result also confirms the weak interaction energy among bright exciton species in TMDs[10]. Surprisingly, dark exciton complexes show a remarkable blueshift as a function of the distance, indicating that they

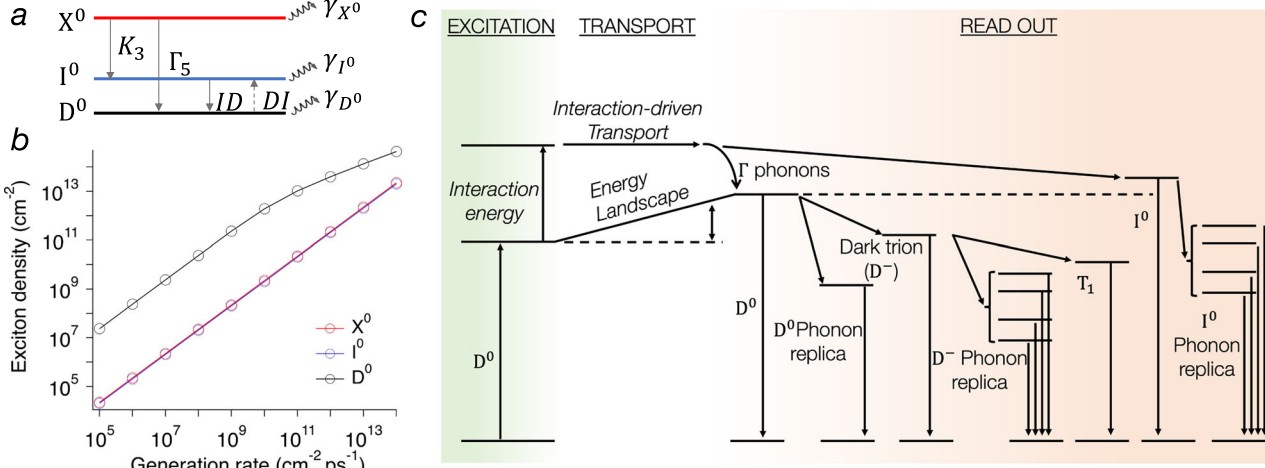

**Fig. 5 | Dark exciton interaction and relaxation paths. a** Scheme of the energy levels and possible transitions among different exciton species. The transition *DI* shown with dash lines indicates the scattering $D^0 \rightarrow I^0$ that occurs only for high values of exciton kinetic energy. **b** Calculated exciton density at the steady state as a function of the generation rate for the excitonic states illustrated in (**a**). **c** The strong interaction of $D^0$ at the excitation position provides initial momentum to propagate uphill. The transported hot dark excitons can then thermalize toward the bottom of their energy dispersion via scattering with low energy $\Gamma$ acoustic phonons. From this state, they can radiatively recombine in the form of a dark exciton $D^0$, dark trions $D^-$, phonon replica $D_p^-$, $T_1$, or $I^0$.

overcome the potential gradient. Dark excitons can diffuse away from the excitation spot, relax to the energy landscape—which represents the lowest energy state available to them at that particular position—and then recombine radiatively. This observation together with the increase of the diffusion length as a function of pump power indicates the buildup of interaction energy in the dark exciton population. This effect is particularly evident for $I^0$, $D^0$, $T_1$, $D_{\Gamma_5}^-$, and $D_{K_3}^-$. This experiment sets a lower limit for the blueshift induced by repulsive interaction to 20 meV, limited by the energy landscape size. The interaction-driven propagation of dark excitons is confirmed in similar diffusion experiments performed at lower excitation powers (Fig. S16) and different energy landscapes, including shorter energy ramps (Fig. S17) and almost flat energy landscapes (Fig. S18).

### Dark exciton interaction and relaxation pathways

We estimate the strength of the exciton-exciton interaction with the theoretical model based on Coulomb interactions between excitons described in refs. 42,43. While the direct contribution to the interaction is generally weak and vanishes for small values of center of mass momentum, the exchange contribution is always repulsive and finite[43,44], with an approximate strength of $U_{ex} \sim a_o^2 E_b$, where $E_b$ is the binding energy and $a_o$ is the Bohr radius. Therefore, the main source of interaction is provided by the exchange term which is spin and valley dependent[4]. Bright and dark excitons have comparable binding energy[45,46] and radius[26], and their exchange interaction strength $U_{ex}$ is expected to be similar. Since the overall interaction energy is given by $\Delta E = n_0 U_{ex}$, the longer propagation and the blueshift of dark excitons observed experimentally is attributed to a higher density. To estimate the order of magnitude of the exciton density, we solve a set of coupled rate equations for the population dynamics of bright, intervalley and dark exciton states as illustrated in Fig. 5a. The model accounts for the radiative lifetime[9], nonradiative phonon-mediated interband transitions[47], and exciton–exciton annihilation processes[48]. Upon excitation, the bright exciton $X^0$ can relax towards the $I^0$ or $D^0$ state. The $X^0 \rightarrow I^0$ transition is induced by the interaction with chiral phonons $K_3$ which promote the spin-preserved intervalley scattering of electrons in the conduction bands from K to K' (see Fig. 1f). The $X^0 \rightarrow D^0$ transition is induced by the interaction with chiral phonons $\Gamma_5$ which promote the spin-flipping intravalley scattering of electrons in the conduction bands. In n-doped samples, the transition $I^0 \rightarrow D^0$ is mediated by scattering with free electrons. Figure 5b shows that in a large

range of generation rates ($G$), the density of dark excitons is almost three orders of magnitude larger than the one of the bright and the intervalley excitons, indicating that $\Delta E$ is expected to be greatest for dark excitons, explaining the experimental observation. We note that the effect of exciton-exciton annihilation is only significant for values of generation rates much larger than the ones used in the experiments. We use a power density of $\sim 5 \cdot$ kW cm$^{-2}$ that, for an absorption efficiency of $\sim 5\%$[49], corresponds to a generation rate of $G \sim 2 \cdot 10^9$ cm$^{-2}$ ps$^{-1}$ returning a density of $D^0$ in the order of $8 \cdot 10^{11}$ cm$^{-2}$. For a blueshift of $\Delta E = 20$ meV as in the case of Fig. 4b, c, the corresponding interaction strength is $U_{ex} \sim 2.5 \cdot 10^{-11}$ meV cm$^2$, which is one order of magnitude larger compared to the one measured in bright excitons[50]. This deviation can originate from the different properties between dark and bright excitons, or from the limitations of the model used to estimate the dark exciton density discussed in the SI. However, the calculated density is compatible with the ones extracted from the experimental data when the interaction strength $U_{ex}$ is estimated by using the values of the bright exciton radius and binding energy reported in the literature (more details in the SI)[45,51,52]. The estimated range of experimental values for the dark exciton density is $1 \cdot 10^{12} - 3 \cdot 10^{12}$ cm$^{-2}$. The agreement between the calculated and measured values of the dark exciton density suggests that the long propagation of dark excitons is promoted by strong exciton–exciton interactions due to their high density and long lifetime.

We note that bright single and triplet trions can relax toward the dark trion state via an electron scattering process with a characteristic time similar to the one of the neutral excitons. Therefore, the density of $D^-$ is expected to be similar to the one of $D^0$[47]. However, due to its the smaller binding energy, the interaction energy experienced by $D^-$ should be smaller.

It is worth noting that despite the similar lifetime of $I^0$ and $D^0$, our model returns a density for $I^0$ which is orders of magnitude smaller than the one of $D^0$. While $D^0$ is the lowest exciton state of the system, the dynamics of $I^0$ is governed by the fast $I^0 \rightarrow D^0$ scattering channel that quickly depopulates $I^0$ in favor of $D^0$. Interestingly, in the experiment of Fig. 4c, $I^0$ recombines at higher energy with respect to the excitation location despite its low density and weak interaction. Even though it is energetically unfavorable, the transition $D^0 \rightarrow I^0$ can occur when the kinetic energy of dark excitons is larger than $\sim 20$ meV[47], compatible to the one observed in our experiments. This observation suggests that $I^0$ generates from the relaxation of the transported $D^0$.

Further diffusion measurements over varying energy landscapes and with different excitation powers are reported in Supplementary Fig. S16–S18.

## Discussion

Our polarization measurements (Fig. 1b), consistent with previous works, indicate that dark exciton replicas posses spin-valley coherence[7,23,25]. Although spin-valley polarization cannot be measured directly from $D^0$ and $D^-$ due to their out-of-plane optical transition, it can be observed in their phonon replicas that reveal the valley information of the original dark states. Therefore, due to their robust diffusion over several micrometers, dark excitons are capable of transporting spin information across large areas and inhomogeneous landscapes. This information can then be read out in the far-field via out-of-plane emission generated by phonon or electron interaction. The transport and relaxation pathways for the dark exciton $D^0$ are schematically illustrated in Fig. 5c. The system is initially excited to create dark excitons as shown in Fig. 5a. The high density of dark excitons favors the strong exciton–exciton repulsive interaction that increases the overall energy of the dark exciton population at the excitation position. This repulsive interaction at the excitation position provides initial momentum to excitons which diffuse away even towards locations of the sample with higher energy landscape. The transported hot dark excitons can then thermalize toward the bottom of their energy dispersion via scattering with low energy Γ acoustic phonons[28]. From this state they can radiatively recombine in the form of dark excitons $D^0$, dark trions $D^-$, phonon replica, $T_1$ or $I^0$.

In summary, we have reported the observation of a robust drift-diffusion process in dark excitons in $WS_2$ that extends over several micrometers in an inhomogeneous energy landscape. We provide evidence that the diffusion is driven by repulsion, stemming from the high exciton density. Due to the introduction of an advanced fabrication process and imaging setup, we are able to study diffusion of dark excitons in a linear potential energy gradient. With these experiments, we show that dark excitons are promising transport means for spin-valley information for applications in quantum information technology.

## Methods

### Sample preparation

2D materials are mechanically exfoliated using a standard scotch tape method on a Polydimethylsiloxane (PMDS) stamp (X4 WF Film from Gel-Pak) and then transferred onto a Si/SiO$_2$ substrate. We use hBN flakes of thickness around $\sim 20$ nm as our bottom and top protective layers. Monolayers of $WS_2$ are initially exfoliated from the bulk crystals (from HQ Graphene) and then transferred onto the hBN substrate. During this process, strain is imprinted on the monolayer by using a large contact angle between the $WS_2$ and the hBN layer. Finally, the 2D material stack is encapsulated with the top hBN layer and the resulting heterostructure is baked at 200° for 5 min in air to remove nanobubbles created at the $WS_2$/hBN interfaces during the transfer process. More details in the SI.

### PL and hyperspectral measurement

PL and spectroscopy measurements are carried out in a home-built confocal microscope setup coupled to a closed-cycle cryostat. PL experiments are performed by exciting the samples non-resonantly with a continuous-wave green laser (532 nm). The laser spot size on the sample is 2 μm. PL maps are taken with a galvanometer mirror scanner in a 4f configuration. The diffraction-limited spatial resolution is approximately 350 nm. An objective lens with NA = 0.9 and free-space-coupled avalanche photodiodes are used for high-efficiency collection. In detection, the excitation laser is filtered out with a 550-nm long-pass filter. Spectra are obtained by directing the signal to a spectrometer with gratings of 600 G/mm. Images of the exciton clouds are taken with an EMCCD camera in the low noise mode. The complete scheme of the optical setup is in Fig. S1.

## Data availability

The datasets generated during and/or analyzed during the current study are available from the corresponding authors on request.

## Code availability

The codes used to generate the data are available from the corresponding authors upon request.

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

## Acknowledgements

G.G. acknowledges support from the National Science Foundation (NSF) (grant No. DMR-2044281), support from the physics department of the Graduate Center of CUNY and the Advanced Science Research Center through the start-up grant, and support from the Research Foundation through PSC-CUNY award 64510-00 52. A.A. and J.Q. were supported through the Simons Foundation and the Air Force Office of Scientific Research. K.W. and T.T. acknowledge support from the Elemental Strategy Initiative conducted by the MEXT, Japan (grant No. JPMXP0112101001) and JSPS KAKENHI (grant No. 19H05790, 20H00354, and 21H05233).

## Author contributions

S.B.C. and G.G. conceived the idea/concept and defined the experimental and theoretical work. S.B.C. prepared samples and performed experimental measurements with the assistance of J.M.W. and J.Q. E.M. implemented the control software for data collection. S.B.C. performed DFT simulations and theoretical calculations. G.G. and S.B.C. analyzed the data. T.T. and K.W. grew hBN samples. G.G. supervised the project. All authors discussed the results and commented on the paper.

## Competing interests

The authors declare no competing interests.
