## [Peer Review File · Nature Communications]

Reviewers' Comments:

Reviewer #1:

Remarks to the Author:

The manuscript by Chand et al investigates how spin-forbidden dark excitons diffuse and interact with each other in the two-dimensional semiconductor WS₂. The authors use a high-quality WS₂ device and a confocal setup with two galvanometer mirrors to show that dark excitons propagate through the flake for longer distances than bright excitons. They use phonon replicas of the dark trion and the semi-dark trion to probe drift-diffusion dynamics, because signals from these states are more readily observable in the far field compared to emission from spin-forbidden dark excitons. By performing laser-power dependent measurements of both the energy of the peaks and the diffusion profile of the dark trion replicas, the authors conclude that there are strong interactions between the dark exciton states causing a further increase in the length of the exciton transport. Lastly, they create a strain profile in the device that causes a potential energy profile and study the diffusion of dark excitons across both increasing and decreasing energy landscapes.

This work builds upon previous reports of spin-forbidden dark exciton diffusion (for example Cadiz et al. Appl. Phys. Lett. 112, 152106 (2018) and Gelly et al. Nat. Comm. 13, 232 (2022), which should both be cited), providing an in depth study of the dark exciton transport properties and interactions. As such, it will be of interest to the broader community and readership of Nature Communications. Before I can recommend it for publication, however, the authors should address the points outlined below.

The authors use the bright exciton diffusion pattern to compare it with the longer-range dark-exciton diffusion. Clearly dark excitons are able to diffuse for longer distances, owing in part to their longer lifetimes, however my main concern is that the diffusion cloud of the bright exciton is very large. Considering that these measurements are taken at 7K, and that bright exciton lifetimes are on the order of 1 ps, a motion of ~ 1 μm (as in figure 2f) would imply velocities on the order of $1\text{e}6$ m/s, which is much larger than reasonable. Of course, this is just a rough estimate, but as the authors can also see in Cadiz et al. APL 112, 152106 (2018), the bright exciton cloud at cryogenic temperatures should match closely with the laser profile. The authors should address this issue in the revised manuscript, either with an analysis of the diffusion data that takes into consideration any experimental artifacts, or with new data. In particular, this is important in order to obtain a proper quantitative understanding of the diffusion profiles of the dark exciton states.

My second concern is regarding the exciton-exciton interaction analysis. The authors perform spectrally resolved photoluminescence measurements of the bright and dark excitonic complexes. They note that the bright exciton states redshift with increasing laser power, while the phonon replicas of the dark trions do not shift at all, due to their interactions. The authors assume, when comparing dark and bright states, that bandgap renormalization effects and coulomb screening are the same for the bright and dark excitons - they should explain in the manuscript why that is the case. Moreover, they should describe the details of their laser excitation protocol, including the frequency and duty cycle of the chopper, to confirm that there is indeed no laser induced heating, since this also causes a redshift in the bright exciton features, and thermalization timescales can be fast in TMDs.

In the model of dark exciton interactions, the authors assume that dark excitons have a permanent out of plane dipole moment of $d \sim 0.5\text{nm}$ that results in dipole-dipole interactions V_{pp} . While the optical transition dipole of the dark exciton transition is indeed in the z direction, this does not necessarily imply that these excitons have a permanent out-of-plane electric dipole moment, and that it is as large as 0.5nm. Early in the text, the authors state: "[...] their permanent out-of-plane dipole - analogous to the one of interlayer excitons in van der Waal heterostructures - enables the control of their spectral properties by an external electric field¹⁴" referencing Zhou et al Nat. Nano (2017). Zhou et al show an out of plane optical transition dipole, but do not show an out of plane electric dipole: there is no evidence of a Stark shift of the dark exciton peak due to an applied electric field. The $d \sim 0.5\text{nm}$, which is larger than the thickness of a single layer TMD, is obtained from Reference 23, a theory paper by Li et al, PRB (2022), which calculates the size of the electron and hole wavefunctions in the z direction, which to lowest order have a gaussian shape of size 0.5nm. This is not the electron hole separation in the z direction in

the simplified dipole picture. The authors should demonstrate that dark excitons have an out of plane permanent electric dipole moment, either with original experimental data, or by citing previous experimental works. In addition, they should provide more evidence that $d \sim 0.5 \text{ nm}$. If not, the authors should not include this dipole-dipole term in the analysis of dark exciton interactions, since in any case the exchange energy contribution is the dominant one. Can the authors also provide a more quantitative comparison of the exciton-exciton interaction strength of dark and bright excitons, either from their data or comparing to previous literature? How does this estimate also compare with the data obtained from the power dependence of the spectra in figure 3? What is the diffusion constant if they use the dark exciton lifetimes provided in the literature?

Finally, some additional minor comments:

-The authors state at the beginning of the manuscript that: “[...] bright excitons in which the electron-hole Coulomb bound pair is formed by an electron and a hole with parallel spin at the K (or K’) valley”, and that dark excitons have electrons and holes with antiparallel spin directions. While it is certainly true that the bright transition is between electron bands with the same spin orientation and the dark transition is between bands with opposite spin orientation, the above statement might cause some confusion given the convention of holes having the opposite spin as the electron band they are in (see for example: Yu et al Arxiv 1507.08103 or Tang et al Nature Comm 10, 4047 (2019))

-The spectrum in figure 1 has many peaks, it would be useful to include a zoomed in version in the supplementary where the main peaks (dark exciton, dark trion, intervalley dark exciton, and some phonon replicas are highlighted). How do the authors select the energy of these peaks? If a fitting is performed, it should be included.

-The authors should explain why D0 seems to be showing very strong chiral emission, given that it should be largely suppressed (for example in He et al Nature Comm 11, 618 (2020)).

-What laser power was used in figure 2b-e? Was it CW or chopped? This should be included more clearly in the text or caption.

-Can the authors explain more clearly why in figure 4 and S8 some of the bright exciton peaks are just as prominent away from the excitation spot as on the excitation spot? Can lower powers or a more focused collection path reduce this issue?

Reviewer #2:

Remarks to the Author:

The authors present a (density-dependent) study of energy as well as space-resolved time-integrated signal in transition metal dichalcogenides monolayers. In particular they find three different behaviours between spin-dark and bright states: i) only the bright states show a redshift in energy with increasing density and ii) only the dark states show an increased diffusion length with increasing density iii) only the dark excitons are able to diffuse uphill. The author attribute these effects to a vertical spatial separation between electron and hole of 0.5 nm for the only dark states. While the results are interesting, the conclusions for the moment are not enough supported by the evidence and in general the analysis shows still too many flaws to deserve publication in Nature Communications. In particular the authors should

- strongly discuss the relationship between electron-hole spatial separation and the so-called optical dipole moment, in particular presenting literature clearly connecting these two quantities for the case of spin-dark excitons in monolayers. In addition comparison to the case of vertical heterostructures should be presented in relation with the polarization displayed by interlayer excitons.

- The authors state that the absence of redshift for dark states is due to the presence of a blueshift canceling the redshift (present for bright states). While it is clear to me that spatially-separated electron-hole would present such a blue-shift in the presence of electric field, it is not equally clear that this has already been observed for the spin-dark states in monolayers. The authors should clearly present literature about this.

- Concerning the density-dependent increase of the diffusion (length) shown for example in Fig. 3f: Can the authors distinguish between an increased diffusion coefficient and/or an increased recombination time?
- Where does the value of 0.5 nm as a dipole come from? It seems very similar to the values found for interlayer excitons in heterostructures; however I would have expected a much smaller value for monolayer excitons.
- The density-dependent blueshift as well as increase of the diffusion length should be quantitatively compared to those found in literature for dipolar interlayer excitons.
- When discussing the uphill diffusion, the authors write "clearly indicates a large repulsion energy that allows them to overcome the potential gradient". This statement should be supported by density-dependent studies.

Reviewer #3:

Remarks to the Author:

The authors report on interaction driven transport of dark excitons at the K point of a WS₂ monolayer at low temperatures (7K).

In a first step they identify the different excitonic features in their PL spectra by comparing with literature, identifying dark and bright excitons. Secondly, the authors investigated the diffusion behavior of the different exciton species. While the bright excitons show low diffusion, the dark excitons spread over the whole sample area. The authors attribute the large diffusion of dark excitons to their long lifetime and exciton-exciton interaction induced by the permanent dipole which generates a drift potential. To demonstrate the strong interaction of the dark excitons power dependent diffusion and spectral measurements were performed. While the bright excitons show a redshift due to band normalization and a reduction of the effective bandgap with increasing power, the dark excitons show no/less redshift, indicating a blueshift generated by the exciton-exciton interaction. In addition, the diffusion spot increases for the dark excitons with increasing laser power.

After this the authors turn to the main claim of the paper the investigation of exciton diffusion in a varying energy landscape.

- 1) excitation and emission on the same spot: All excitons show a redshift of 20meV due to the increasing strain potential over 6 micrometer, leading to an maximum strain of 0.4%
- 2) excitation in the low strain region and detection for increasing strain values (energy downhill): The bright neutral excitons shows a redshift due to the funneling effect, same holds for the dark excitons where in addition to the funneling also repulsion interaction plays a role.
- 3) excitation at high strain and detection at various lower strain values (energy uphill): all bright excitons show no energy shift. In contrast the dark excitons show a strong blueshift indicating that they diffuse uphill and recombine at a position with lower strain (higher energy). The authors conclude that the high repulsion energy allows the dark excitons to overcome the potential gradient. They claim that the different diffusion behavior is due to the weak interaction of the in-plane dipoles of the bright excitons compared to the strong interaction of dark excitons with out-of plane dipoles.

From the uphill diffusion the authors estimate a repulsive interaction of 20meV from which they calculated a dark exciton density of $2.5 \cdot 10^{11} \text{ cm}^{-2}$.

In my opinion the paper is reporting on an interesting finding, however the explanation is in the current form not very convincing to me. In addition the argumentation is somehow confusing for me.

Therefore, I do not recommend publication in Nature Communications in the current form.

In the following I have several detailed comments where in my opinion the study is not convincing:

- In Fig. 2 the authors show the diffusion spots of dark and bright excitons. For the dark excitons there seems to be even emission visible next to the flake. Why is that the case? The white outlines of the monolayer look very different to the image in a. What is the reason?

- The diffusion length of the dark excitons (Fig 2 d, e and Fig 3 d) seems to be limited by the flake edges. Have the authors measured on a larger flake, and can they estimate the diffusion length for the different species?

- Fig 3 a. For me it is surprising that the blueshift due to exciton-exciton interaction of the dark excitons exactly compensates for the redshift of the bandgap reduction. Do the authors have an explanation for that? Can the authors show the fitting procedure for the spectra and give an error estimation for their extracted exciton energies?

- The authors put the energetic blue shift of the dark excitons with increasing excitation power directly in line with their larger diffusion. However, this relation is not directly clear to me. Can they explain in more detail and give references supporting this?

- Where the measurements presented in Figures 1-3 done on the same flake as the measurements in Figure 4 (strain landscape)? If so, why is the strain gradient not affecting the diffusion spots in Figures 1-3? I would expect a distorted diffusion spot due to the funneling effect. Especially for the neutral excitons, the authors show that the excitons move downhill but not uphill, therefore one would expect an asymmetric diffusion spot on a sample with a spatial strain gradient. In case the measurements in Figs. 1-3 were done on a different sample (without strain gradient) the authors could repeat the diffusion measurement on the sample of Fig. 4. Do they see a distorted diffusion?

- The authors should provide a strain map of the whole sample (which can be easily done by mapping the bright exciton energy as a function of position). The authors only show the spectra and strain profile along a line (Fig4), but for the diffusion the strain landscape in 2D is important. It is also not clear where the measurements are performed on the sample. The authors should indicate the positions for excitation and emission on the sample. I assume the authors measure from one edge of the flake to the other? However, the presence of an edge would be important in the discussion of diffusion directions.

- The authors claim that the reason they detect the unshifted bright exciton, trion, and biexciton emission a few micrometers away from the excitation spot (Fig 4b, c) is due to their high NA and that they still pick up the emission from approx. 2.5 μm away (page 5 and 6). I assume there is a strain gradient even in the emission area of the diffusion spot (which has a size of about 2 microns). Do they see any line broadening due to the summation of different strain values? Can the authors estimate the strain variation in the diffusion spot of the bright excitons? I wonder why they do not see broad line shapes for the dark excitons, which have a much longer diffusion length. In this case the large detection spot of more than 2.5 micrometer will cover emission areas of different strain values.

- In figure 4 the authors excited at a low strain level and measured for increasing strain and the other way around. The authors should also excite at an intermediate strain level and measure spectra at lower and higher strain positions on the sample. In this case they can directly measure up and downhill diffusion in the same excitation scenario.

- In figure 4 the colored dots used to mark the excitons are hardly visible and make it difficult to match the spectral feature with the correct exciton species.

- The authors explain their results due to the repulsion driven propagation of dark excitons caused by the strong interaction due to the permanent out-of-plane dipole (Page 4 Line 7). Can the authors explain this in more detail? For me it is not clear why the out-of-plane dipole of dark excitons should create a stronger exciton-exciton interaction as the in-plane dipole of the bright excitons. Can they give references?

What is actually the assumed mechanism for the diffusion of the bright excitons if it is not exciton-exciton interaction? Nothing is mentioned in the manuscript.

- The authors write "...the diffusion of D- is expected to be limited by the smaller binding energy of the exciton carrier complex." (page 6) This connection is not clear to me. Why does the diffusion depends on the binding energy? Can the authors give references.

In addition on page 5 they write "Despite its short lifetime X0 has a large binding energy and tensile strain generates a funneling effect...". It is true that X0 has a large binding energy but how is it connected to the diffusion?

- Can the authors perform lifetime measurements to estimate the diffusion speed for the different exciton types?

- Page 6: what is meant by 'relax to their ground state' this state is not well defined in the text.

- What is the point of estimating the dark exciton density? This estimation appears rather unmotivated to me in the text. Is their result in line with their expectation, and how does it differ for the bright exciton density?

- The authors claim that they used a new fabrication technique in order to produce a sample with a strain gradient. This is not true. In contrast such strain landscapes appear often naturally in transferred layers since a contact angle is often unavoidable. Samples with large strain gradients were already used before (Nature Communications 12, 7221 (2021))

- In their introduction the authors mention dynamics of interlayer excitons. However, citations regarding the (strain) dependent excitons diffusion of momentum dark excitons are missing. E.g. Nature Communications 12, 7221 (2021)

- The schematic figure 1a can lead to misunderstandings. Why are the dark excitons depicted above the monolayer floating in the air?

General remarks for all Reviewers

We would like to thank all the Reviewers for carefully reviewing our manuscript and for raising illuminating comments and questions that have helped to improve the overall quality of our paper. Fruitful comments from multiple Reviewers have highlighted the presence of a few unclear passages of our previous manuscript. Prompted by these comments, we have improved the analysis of our data and developed a model that better captures the physics of dark exciton interactions. Before detailing a point-to-point answer to the individual reviewers, we summarize here important general remarks that represent the major changes from the previous version of the manuscript. In our response, we use different fonts to differentiate between the original Reviewers' reports and our answers. We highlight in blue color the changes to the manuscript.

RECONSIDERATION OF DIPOLE-DIPOLE INTERACTION

To understand the experimental observation of long-range transport of dark excitons, we presented in our initial paper a simple analysis of exciton interactions based on two terms: dipole and exchange interactions. As correctly pointed out by the Reviewers, this model for exciton interactions implies the presence of a permanent dipole for dark excitons, which is not completely ascertained. Moreover, in a wide range of exchange wave vectors, the interactions of excitons is dominated by exchange terms, while the direct dipole interaction, if present, is smaller. Therefore, the discussion on the permanent dipole is not essential to explain our experimental findings. Prompted by the Reviewers' comments, we update the theoretical model for exciton-exciton interaction in dark excitons to consider only exchange interactions and not dipole-dipole interactions. We would like to note that in the new version of the manuscript we compare this model with the experimental results finding good agreement.

The predominance of exchange interaction over direct one was established for quantum well excitons by C Ciuti et al, (PRB 58, 7926 (1998)), and recently confirmed in TMDs by D. Erkensten et al., (PRB 103, 045426 (2021)). We note that for dark excitons in TMD monolayers, exchange interaction is the dominant term even in the possible presence of dipole-dipole interaction. In fact, as a rule of thumb, while dipole interaction scales approximately as $U_{dd} \sim da_0 E_b$, exchange interaction scales as $U_{ex} \sim a_0^2 E_b$, where d is the exciton dipole moment, a_0 is the Bohr radius, and E_b the binding energy. Since the vertical dipole d in a monolayer is limited by the monolayer thickness (~ 0.3 nm), it is always almost an order of magnitude smaller than a_0 (~ 2 nm). Therefore, the exchange interaction is the dominant term.

To better describe the type of interaction in the manuscript, we remove all the previous references to permanent dipole and the final discussion on exciton-exciton interactions. We add the following paragraph along with references to D. Erkensten et al., PRB 103, 045426 (2021) and C Ciuti et al, PRB 58, 7926 (1998).

“We estimate the strength of the exciton-exciton interaction with the theoretical model based on Coulomb interactions between excitons described in Ref. ^{42,43}. While the direct contribution to the interaction vanishes for small values of center of mass momentum, the exchange contribution

is always repulsive and finite,^{43,44} with an approximate strength of $U_{ex} \sim a_o^2 E_b$, where E_b is the binding energy and a_o is the Bohr radius. Therefore, the main source of interaction is provided by the exchange term which is spin and valley dependent.⁴ »

NEW CALCULATIONS OF EXCITON DENSITY

In the new model we consider as predominant the exchange interaction. Therefore, the strength of exchange interactions ($U_{ex} \sim a_o^2 E_b$) of bright and dark excitons is expected to be similar as they have a similar Bohr radius (a_o) and binding energy (E_b). Therefore, observed differences in the dynamics of bright and dark excitons in the experiments are attributed to the density of the exciton populations since the effective interaction energy is given by $\Delta E = n_o U_{ex}$. **In the revised manuscript, we include new calculations to estimate the exciton density for bright, intervalley and dark excitons. The value of the dark exciton density calculated with this model is in good agreement with the one extracted from our experimental value of the interaction energy when considering exchange interactions.**

To estimate the order of magnitude of the exciton density, we develop a model based on coupled rate equations that accounts for radiative and non-radiative decay pathways, as well as exciton annihilation. At the steady state, this calculation returns a significantly higher density of dark excitons. As a result, the effective interaction potential energy ($\Delta E = n_o U_{ex}$) is greater for dark excitons and in agreement with our experimental data. In this model, we consider natural bright (X_o), intervalley (I_o) and dark excitons (D_o). We take into account their energetic ordering ($E_{X_o} > E_{I_o} > E_{D_o}$), the phonon-mediated scattering processes among different bands, and the different lifetimes. For the scattering rates among excitonic states, we use the theoretical values provided by the recent calculations performed for WSe₂ at T = 4K by M. Yang et al., [PRB 105, 085302 (2022)]. Due to the similarities between the two materials, we expect these values to not change significantly in WS₂.

To the best of our knowledge, our manuscript is the first report on the experimental observation of intervalley excitons in WS₂ and there are no measurements on their radiative lifetime, yet. In the model, we use the lifetimes reported for WSe₂ in Ref. by Z. Li et al. [ACS Nano 13, 14107 (2019)] in which they have been measured lifetime consistently for bright, dark and intervalley excitons at cryogenic temperature. We note that the lifetime in WS₂ for bright and dark excitons has the same order of magnitude as WSe₂, and we expect the same for the intervalley. Moreover, at the steady state, the dynamics is dominated by interband scattering rates (in the order of a few ps^{-1}) and small variations of the radiative lifetime (in the order of hundred of ps) do not significantly affect the overall behavior of exciton populations.

In the calculation, we also consider nonradiative exciton-exciton annihilation (EAA) processes to include any possible Auger recombination that could happen at high exciton density. However, this process is strongly suppressed in encapsulated samples [PHYSICAL REVIEW B 95, 241403(R) (2017)], and the effect of this annihilation process emerges only at very high density.

The results of the calculations are shown in the figure below. At the steady state, the density of dark excitons is almost three orders of magnitude larger than the one of the bright and the intervalley excitons. This result shows that the overall interaction energy $\Delta E = n_0 U_{ex}$ is expected to be the largest for dark excitons, in agreement with the experimental observation.

Figure R1: Calculations of the exciton density. **a** - Schematic of the excitonic states, the radiative and non-radiative decay pathways. **b** - Exciton density at the steady state as a function of the generation rate.

To validate our model we compare the order of magnitude of the calculated dark exciton density with the one extracted from the experimental data. We perform the density calculations over a range of generation rates compatible with the one used in the experiments. See Figure 1b above. In our experiments, (Figure 4 of the main text), we use a generation rate of $G \sim 7 \cdot 10^8 \text{ cm}^{-2} \text{ ps}^{-1}$. With this value, we calculate a density of D_0 of the order of $\sim 2 \cdot 10^{11}$. This value is compatible with the one extracted from the experimental data of Figure 4 of the main text in which we measured a dark exciton interaction energy of $\Delta E = 20 \text{ meV}$. With this value of ΔE and using $E_b = 0.7 \text{ eV}$ and $a_0 = 1.8 \text{ nm}$, we expect a dark exciton density of $\sim 6 \cdot 10^{11} \text{ cm}^{-2}$. Therefore, we observe a good agreement between the calculated and measured values of the dark exciton density, corroborating the hypothesis that the long propagation of dark excitons is promoted by strong exciton-exciton interactions due to high density.

We note that the theory developed by M. Yang et al., [PRB 105, 085302 (2022)] predicts the process $D_0 \rightarrow I_0$ for large values of exciton kinetic energy. This process is compatible with the observation of I_0 excitons at large distances from the excitation spots, as shown in Fig. 4c. Although we are aware that more sophisticated models should be developed to capture the full dynamics of carriers in W-based monolayers, we note that this simple model already qualitatively explains our results. In fact, the inclusion of trions is not expected to change qualitatively our conclusions as the relaxation processes of bright trionic states mostly occur

among dark trionic states and have similar rates as the one of neutral excitons [PRB 105, 085302 (2022)]. Moreover, we note that the finite density of chiral phonons (not included in the model) could lead to different population dynamics at high generation rates. In the model, we neglect the relaxation path towards the lower-energy momentum-forbidden $K\Lambda$ exciton because this interband scattering process in WS^2 has been found to be significant only in the presence of compressive strain [Feierabend, et al., Phys. Rev. B 99, 195454 (2019), Chand, et al. Nano Lett. 22, 3087(2022).].

We include all the details of the model in the SI and the discussion on the results of the new calculations in the manuscript. In the main text, we add a new figure (Fig.5) to illustrate the results of the density calculator and better visualize the relaxation process far away from the excitation. Moreover, we add the references to Z. Ye et al., Nature 513, 214 (2014), Robert et al., Nat. Comm. 11, 4037 (2020), Y. Li et al., PRB 90, 205422 (2014), Hoshi et al., Phys. Rev. B 95, 241403(R) (2017), Yang et al., Phys. Rev. B 105, 085302 (2022).

When discussing exciton-exciton interactions, we add:

“Bright and dark excitons have comparable binding energy^{45,46} and radius,²⁶ and their exchange interaction strength U_{ex} is expected to be similar. Since the overall interaction energy is given by $\Delta E = n_0 U_{ex}$, the longer propagation and the blueshift of dark excitons observed experimentally is attributed to a higher density. To estimate the order of magnitude of the exciton density, we solve a set of coupled rate equations for the population dynamics of bright, intervalley and dark exciton states as illustrated in Fig. 5a. The model accounts for the radiative lifetime,⁹ nonradiative phonon-mediated interband transitions,⁴⁷ and exciton-exciton annihilation processes.⁴⁸ Upon excitation, the bright exciton X^0 can relax towards the I^0 or D^0 state. The $X^0 \rightarrow I^0$ transition is induced by the interaction with chiral phonons K_3 which promote the spin-preserved intervalley scattering of electrons in the conduction bands from K to K' (see Fig1f). The $X^0 \rightarrow D^0$ transition is induced by the interaction with chiral phonons Γ_5 which promote the spin-flipping intravalley scattering of electrons in the conduction bands. In n-doped samples, the transition $I^0 \rightarrow D^0$ is mediated by scattering with free electrons. Fig. 5b shows that in a large range of generation rates (G), the density of dark excitons is almost three orders of magnitude larger than the one of the bright and the intervalley excitons, indicating that ΔE is expected to be greatest for dark excitons, explaining the experimental observation. Moreover, we find agreement with the dark exciton density from the energy blueshift measured in Figure 4b,c. We use a power density of $\sim 5 \cdot kW \cdot cm^{-2}$ that, for an absorption efficiency of $\sim 5\%$,⁴⁹ corresponds to a generation rate of $G \sim 7 \cdot 10^8 \cdot cm^{-2} \cdot ps^{-1}$ returning a density of D^0 in the order of $2 \cdot 10^{11} \cdot cm^{-2}$. This value is compatible with the one $\sim 6 \cdot 10^{11} \cdot cm^{-2}$ extracted from the experimental data of Fig. 4c in which we measure a dark exciton interaction energy $\Delta E = 20$ meV. The agreement between the calculated and measured values of the dark exciton density suggests that the long

propagation of dark excitons is promoted by strong exciton-exciton interactions due to their high density and long lifetime.

We note that bright single and triplet trions can relax toward the dark trion state via an electron scattering process with a characteristic time similar to the one of the neutral excitons. Therefore, the density of D^- is expected to be similar to the one of D^0 .⁴⁷ However, due to its the smaller binding energy, the interaction energy of D^- should be smaller. ”

In the SI, we remove the old description of exciton-exciton interaction. In the section 6 of the SI entitled “Exciton-exciton interaction and exciton density” we add all the details of the calculations for the exciton density and for the estimate of the generation rate as reported above.

In the main text, we add a discussion regarding the propagation of I_0 : “Interestingly, in the experiment of Fig.4c, I^0 recombines at higher energy with respect to the excitation location despite its low density. Even though it is energetically unfavorable, the transition $D^0 \rightarrow I^0$ can occur when the kinetic energy of dark excitons is larger than ~ 20 meV,⁴⁷ compatible to the one observed in our experiments. This observation suggests that I^0 generates from the relaxation of the transported D^0 .”

MEASUREMENT OF THE DIFFUSION LENGTH

Prompted by the Reviewers’ questions on the estimation of the diffusion length of bright and dark excitons, we carry out a more careful analysis of our data and make significant updates to Figures 2 and 3 that describe the expansion of the exciton clouds. In particular, we perform new measurements to confirm the size of our excitation spot to compare with the size of the exciton clouds. Moreover, we provide a more detailed characterization of the strain landscape and we analyze the expansion of the exciton clouds across strained and unstrained regions. This analysis reveals an asymmetry in the exciton cloud expansion due to the uphill energy landscape generated in the strained regions. This asymmetry is confirmed by the measurement of the diffusion length that we perform as discussed below.

To answer the reviewers’ questions, we have estimated crucial parameters to quantify exciton transport. We calculate the diffusion length L_D using a two-dimensional diffusion model with a point source that has been proven to be reliable for 2D excitons. We adopt the approach used by L.A. Jauregui et al. [Science 366, 870 (2019)] to analyze the emission profiles. To quantify the diffusion length, we use the asymptotic value of the steady-state density profile given by $n(r) \sim e^{-r/L_D} \sqrt{r/L_D}$ to fit the emission profile of the different exciton clouds away from the laser spot. The diffusion lengths L_D as a function of the excitation power are now shown in Fig. 3f. We observe that the diffusion length of dark excitons increases with the excitation power, while the

diffusion length of bright excitons remains constant. This observation confirms the result on the expansion of the exciton cloud reported in the previous version of the manuscript, and that the diffusion of dark excitons is driven by repulsive interactions. Moreover, we observed a strong asymmetry between the diffusion length of dark excitons in flat landscape (unstrained) and the diffusion length in an uphill landscape (strained). This asymmetry is consistent with the diffusion measurements reported in Fig. 4 of the main text. The results of these new measurements are shown below in Figure R2.

Figure R2: The diffusion length of bright and dark excitons in strained uphill energy and unstrained flat energy direction. The error bars are the error of the fit.

We obtain a diffusion length for bright excitons of approximately $1 \mu\text{m}$ which is surprisingly high. This high value is attributed to the contribution of a slower decay rate observed in excitons in TMDs. Time-dependent photoluminescence experiments have observed two mechanisms for exciton radiative decay: an initial rapid decay (τ_1) followed by a slower decay (τ_2). This has been observed and discussed for bright excitons by G. Wang et al., PRB 90, 075413 (2014), and for interlayer excitons by L. A. Jauregui et al., Science 366, 870 (2019). The slower decay rate τ_2 of bright excitons in WSe_2 reported by G. Wang et al. is much longer than the radiative lifetime τ_1 . We note that, since we measure the exciton cloud in the steady-state regime, we expect a rapid decay rate at the center of the excitation and a significantly slower decay rate away from the center of excitation. Therefore, in a steady-state condition, the diffusion of bright excitons is influenced by both the rapid and slow decay rates. The latter contributes to a longer diffusion length. This observation is supported by the results in WSe_2 by Cadiz et al. APL 112, 152106 (2018), where the authors reported a diffusion length of short decay ($L_1 = 0.2 \pm 0.05 \mu\text{m}$) followed by a longer one ($L_2 = 1.5 \pm 0.04 \mu\text{m}$). Therefore, our measurements of diffusion length of

~1 μm for bright excitons is consistent with these previous findings. We also note that, to the best of our knowledge, the origin of this slow decay is still under debate. Most importantly for this work, previous works show that dark excitons possess only one slow decay time (see for example Cadiz et al. APL 112, 152106 (2018) and Z. Li et al. ACS Nano 13, 14107 (2019)). From L_D , we can estimate the diffusion constant according to $D = L_D^2/\tau$. For dark excitons we obtain a power-dependent diffusion constant ranging from 40 to 240 cm^2/s which is in agreement with the previous report of Cadiz et al.

To account for these updates, we make a series of modifications to the manuscript. All the technical aspects related to the calculator of the diffusion length and constant are now discussed in detail in the supplementary information.

In the main text, we add the references Cadiz, F. et al., J. Appl. Phys. 116, 023711 (2014), Cadiz, F. et al. Appl. Phys. Lett. 112, 152106 (2018), and Wang, G. et al. Phys. Rev. B 90, 075413 (2014). When discussing diffusion length, we add:

“In addition, we estimate the characteristic diffusion length of the dark excitons by fitting the line profiles away from the center of laser excitation with the two-dimensional diffusion model with a point source according to $n(r) \sim e^{-r/L_D} \text{sqrt}(r/L_D)$.^{16,19,35} To investigate the effect of exciton-exciton interaction, we monitor the diffusion length at different excitation powers. Figure 3f shows that L_D increases from 1.4 to 2.4 μm in the unstrained (+x) directions and from 1 to 1.6 μm in the strained (-x) region. The asymmetry in the diffusion length in the strained uphill and unstrained directions is attributed to the different energy landscape. Figure S11 in the SI shows the characterization of strain along the horizontal direction x . After a quick interface region, in the negative x direction, strain creates a smooth and linear uphill energy gradient that limits diffusion. In both directions, the increase of the characteristic diffusion length L_D with increasing excitation power indicates an enhanced diffusion at high exciton densities. Similarly to interlayer excitons,^{16,17} exciton interactions promote longer diffusion. We note how the diffusion length of dark excitons is comparable to the one of interlayer excitons despite the shorter lifetime (τ), suggesting a different interaction strength. As discussed below, we attribute the increase of diffusion length of dark excitons to the large interaction energy. The diffusion constant ($D = L_D^2/\tau$) of dark excitons in the unstrained region ranges from 40 to 240 cm^2/s , in agreement with previous reports.¹⁹ Differently from dark excitons, the diffusion length of X^0 is not density dependent and stays constant around 1 μm for all pump powers. This surprisingly large value for the diffusion length of bright excitons has been already reported¹⁹ and attributed to the combined contribution of a rapid and slower decay rate.³⁶”

We add Note 6 to the SI:

“6. Analysis of exciton diffusion

We calculate the diffusion length L_D using a two-dimensional diffusion model with a point source that has been proven to be reliable for 2D excitons. We adopt the approach used by L. A. Jauregui et al. to analyze the emission profiles.¹⁷ To quantify the diffusion length, we use the asymptotic value of the steady-state density profile given by $n(r) \sim e^{-r/L_D} \sqrt{r/L_D}$ to fit the emission profile of the different exciton clouds away from the laser spot. We use a steady-state width of 2.2 μm , and the extracted diffusion lengths are plotted in Figure 3f of the main text.

Previous time-dependent photoluminescence experiments have observed two mechanisms for intensity decay in TMDs: an initial rapid decay (τ_1) followed by a slower decay (τ_2). This has been observed for bright excitons¹⁸ and interlayer excitons.¹⁷ At this point, there are no reports of a similar behavior in the dark excitons. We note that the value of L_D measured in the steady-state and in low exciton density regime would be dominated by a slower decay rate. Furthermore, the lifetime τ of spin-dark excitons reported in the literature ranges from 110 ps to 250 ps.^{4,19} For the purposes of estimation and consistency with the calculations of exciton density, we use a lifetime of 250 ps and we estimate the diffusion constant according to $D = L_D^2 / \tau$. We find that the diffusion constant (D) ranges from 68 to 240 cm^2/s in the unstrained direction (+x), while it ranges from 40 to 110 cm^2/s in the strained uphill direction (-x). The diffusion constant of dark excitons increases with excitation power due to repulsive interaction among dark excitons.”

In Figure 2 of the main text, we rotate and resize the PL map in Fig.2a to ease the comparison with the measurements of exciton propagation reported in Fig.2c,d,e. In Fig.2a we highlight the different regions of the sample, and the position of the laser spot for the diffusion measurements. We move the new measurements of the laser profile to the SI in Figure S7.

In Figure 3 we update the line profiles of the exciton cloud expansion in the strained uphill and unstrained regions in figures 3e and 3f to better illustrate the asymmetry observed in our experiments. To quantify this asymmetry, we calculate the diffusion length L_D in both directions, and this analysis has been added to figure 3f.

REVIEWER COMMENTS

Reviewer #1 (Remarks to the Author):

The manuscript by Chand et al investigates how spin-forbidden dark excitons diffuse and interact with each other in the two-dimensional semiconductor WS₂. The authors use a high-quality WS₂ device and a confocal setup with two galvanometer mirrors to show that dark excitons propagate through the flake for longer distances than bright excitons. They use phonon replicas of the dark trion and the semi-dark trion to probe drift-diffusion dynamics, because signals from these states are more readily observable in the far field compared to emission from spin-forbidden dark excitons. By performing laser-power dependent measurements of both the energy of the peaks and the diffusion profile of the dark trion replicas, the authors conclude that there are strong interactions between the dark exciton states causing a further increase in the length of the exciton transport. Lastly, they create a strain profile in the device that causes a potential energy profile and study the diffusion of dark excitons across both increasing and decreasing energy landscapes.

This work builds upon previous reports of spin-forbidden dark exciton diffusion (for example Cadiz et al. Appl. Phys. Lett. 112, 152106 (2018) and Gelly et al. Nat. Comm. 13, 232 (2022), which should both be cited), providing an in depth study of the dark exciton transport properties and interactions. As such, it will be of interest to the broader community and readership of Nature Communications. Before I can recommend it for publication, however, the authors should address the points outlined below.

We thank the reviewer for the positive assessment of our work and for highlighting a few important points. We are grateful for the thoughtful reviews of our manuscript and pleased to hear that our work on spin-forbidden dark exciton diffusion in WS₂ is “of interest to the broader community and readership of Nature Communications.”

We acknowledge the previous studies by Cadiz et al. and Gelly et al. on spin-forbidden dark exciton diffusion and agree with the Reviewer that they are important references in this field. We cite them in the revised version (Ref. 19 and 21) and compare our results with theirs. We would also highlight the novelty and significance of our study in the context of the existing literature. Beside observing long diffusion for dark exciton species, in our work we provide evidence of dark exciton interaction and show that dark exciton transport is resilient to disorder-induced energy landscapes. We also elucidate the relaxation pathways of dark excitons that can be used as optical read-out in optoelectronics.

We address all reviewer's specific concerns, follow her/his suggestions and change the manuscript accordingly. We hope that the Reviewer could recommend publication of this new version of the manuscript in Nature Communications. Before answering point-to-point to the comments, we refer the Reviewer to the general remarks at the beginning of our response where we discussed the major updates and changes to the manuscript.

The authors use the bright exciton diffusion pattern to compare it with the longer-range dark-exciton diffusion. Clearly dark excitons are able to diffuse for longer distances, owing in part to their longer lifetimes, however my main concern is that the diffusion cloud of the bright exciton is very large. Considering that these measurements are taken at 7K, and that bright exciton lifetimes are on the order of 1 ps, a motion of $\sim 1 \mu\text{m}$ (as in figure 2f) would imply velocities on the order of $1\text{e}6 \text{ m/s}$, which is much larger than reasonable. Of course, this is just a rough estimate, but as the authors can also see in Cadiz et al. APL 112, 152106 (2018), the bright exciton cloud at cryogenic temperatures should match closely with the laser profile. The authors should address this issue in the revised manuscript, either with an analysis of the diffusion data that takes into consideration any experimental artifacts, or with new data. In particular, this is important in order to obtain a proper quantitative understanding of the diffusion profiles of the dark exciton states.

The Reviewer raises an important concern regarding the diffusion of bright excitons observed in our experiments. We would like to thank her/him for highlighting this point and for stimulating us to double check for potential experimental artifacts.

Prompted by this comment, we check for experimental errors and conduct further analysis of our data. For an in depth discussion and list modifications to the manuscript, we refer the Reviewer to the general remarks at the beginning of our response, in particular the section "MEASUREMENT OF THE DIFFUSION LENGTH". Here we just summarize the main conclusions.

We have confirmed with new measurements the size of our excitation spot to compare to the size of the exciton clouds shown in Fig. 2 of the manuscript. The new measurement of the excitation spot is in Fig.S9a of the SI. Moreover, we extract the values of diffusion length for both neutral excitons and phonon replicas of dark excitons. For the bright exciton we obtain a value of $L_D \sim 1 \mu\text{m}$ which is constant as a function of the pump power. We agree with the Reviewer that this value is surprisingly high and a few comments are in order.

First, the value of $L_D \sim 1 \mu\text{m}$ for bright excitons is in agreement with the paper mentioned by the Reviewer, Cadiz et al. APL 112, 152106 (2018). While discussing the diffusion of bright excitons, Cadiz et al. wrote: "*For the bright neutral exciton, the spatial decay cannot be accounted for with just one diffusion length. By allowing two length scales for X_0 , one extracts a short decay ($L_1 = 0.2 \pm 0.05 \mu\text{m}$) followed by a longer one ($L_2 = 1.5 \pm 0.05 \mu\text{m}$).*" In the analysis of the diffusion length, we consider diffusion far away for the excitation spots, namely we mostly consider the long diffusion term which returns a comparable value with the one of Cadiz et al. The origin of this longer diffusion term is associated with two mechanisms for radiative decay observed in time-dependent photoluminescence experiments. In both bright excitons (G. Wang et al., PRB 90, 075413 (2014)) and interlayer excitons (L.A. Jauregui et al., Science 366, 870–875 (2019)) the initial rapid decay is followed by a much slower decay. The study conducted by G. Wang et al. reported a slower decay rate of bright excitons in WSe_2 of

approximately 33 ps. By considering the slower radiative decay, the diffusion velocity decreases by at least one order of magnitude. We note that at the steady state, the measurements of L_D is dominated by slow decay. We did not find any report in the literature of the slow decay for dark excitons. Therefore, our estimation of the diffusion parameters for dark excitons should not be affected by the slow decay.

We also note that in our measurements, we use an EMCCD with very high dynamic range that allows us to detect very dim light and observe the expansion of bright excitons due to the slow decay.

We appreciate the reviewer's insight and we believe that we now provide a more thorough and accurate analysis of the diffusion profiles observed in our experiments. As a response to this comment we add the following sentence to the main text: “Differently from dark excitons, the diffusion length of X^0 is not density dependent and shows a constant value around 1 μm for all pump powers. This surprisingly large value for the diffusion length of bright excitons has been already reported¹⁹ and attributed to the combined contribution of a rapid and slower decay rate.³⁶”

My second concern is regarding the exciton-exciton interaction analysis. The authors perform spectrally resolved photoluminescence measurements of the bright and dark excitonic complexes. They note that the bright exciton states redshift with increasing laser power, while the phonon replicas of the dark trions do not shift at all, due to their interactions. The authors assume, when comparing dark and bright states, that bandgap renormalization effects and coulomb screening are the same for the bright and dark excitons - they should explain in the manuscript why that is the case. Moreover, they should describe the details of their laser excitation protocol, including the frequency and duty cycle of the chopper, to confirm that there is indeed no laser induced heating, since this also causes a redshift in the bright exciton features, and thermalization timescales can be fast in TMDs.

We thank the Reviewer for these questions regarding the analysis of the interaction energy in our power-dependent measurements. In response to the first question, we initially assumed that the bandgap renormalization effects and Coulomb screening are the same for bright and dark excitons because they originate from the same material and are subject to the same dielectric environmental conditions. Previous studies have shown that these effects are typically material-dependent [Nat Com. 8, 15251 (2017), ACS Nano 11, 12601 (2017), Nano Lett. 14, 3743(2014)]. However, we agree with the Reviewer that this is just an assumption and, to the best of our knowledge, there are no reports of dedicated measurements on the effect of band gap renormalization on specific excitonic complexes.

After a careful analysis and prompted by the questions from other Reviewers, we tone down the discussion on the power-dependent exciton blueshift and move the power dependent spectra to the SI. The reason for this choice is that this piece of data, despite supporting our conclusion, does not allow for a quantitative analysis and does not add much to the discussion on the interaction-driven enhanced transport of dark excitons, which is the main focus of our work. We would like to stress that the increase of the diffusion length as a function of the pump power, and

the uphill diffusion can be quantitatively analyzed and represent clear and more relevant evidence of interaction-driven transport of dark excitons.

The quantitative analysis of the power-dependent spectra is limited by two factors. On one hand, it is challenging to precisely extract the energy shift of dark excitons because the spectrum at high power is fully dominated by charged biexcitons and a precise peak deconvolution is impossible. Second, without precisely knowing how bandgap renormalization affects different exciton complexes we cannot quantitatively compare the energy shift of individual excitons due to interactions. However, we observed a different shift rate for bright and dark exciton species which indicate a different interaction. This shift can be quantified only up to medium laser powers (corresponding to the blue spectrum in Fig. S7). Therefore, we move these data to the SI as Figure S7 and better highlight the different energy shifts.

In the main text, we add the following discussion to clarify the effects and Coulomb screening in power dependent spectra:

“The evolution of the emission spectrum for increasing pump power, reported in Fig. S7 in the SI, shows a smaller redshift due to band renormalization for dark exciton species when compared to the bright counterparts.³²⁻³⁴ This suggests the dark excitons are affected by stronger repulsive interactions.”

We move the detailed discussion regarding the power-dependent spectra to the Note 4 of the SI, together with the experimental data. It reads:

“4. Power-dependent spectral emission

Figure S7 shows the emission spectra taken with increasing laser power, from 0.05 μW to 500 μW . At high excitation power, the spectra are dominated by the strong emission of bright excitons (XX^- , X^0 and X_T^- , X_S^-), and the precise deconvolution the emission energy of D_p^- and T_1 can not be done. As previously observed in similar samples, at high carrier density a redshift of the exciton complexes occurs because of band renormalization and a reduction of the effective bandgap.¹⁴⁻¹⁶ We perform the experiments with a CW laser at a wavelength of 532 nm with a chopper frequency of 500 Hz and a duty cycle of 25%. Therefore, we can exclude major effects due to heating of the sample. As far as D_p^- and T_1 are resolvable in the spectra (blue curve in Fig. S7), it is possible to observe that they experience a smaller energy shift compared to the redshift of bright exciton complexes. The redshift of bright excitons is -0.65 ± 0.05 meV, while the shift of D_p^- is -0.23 ± 0.02 meV. This difference indicates that, at high density, dark excitons undergo different interaction processes. However, we do not know how band gap renormalization affects the different exciton complexes and a quantitative analysis is impossible at this moment.

Figure S7: Emission spectra at different excitation powers ranging from 0.05 to 500 μ W. The emission energy at low power for the neutral excitons X^0 , the negatively charged biexciton XX^- , the dark exciton T_1 and the dark trion phonon replica D_{K3}^- is highlighted with vertical lines. The spectrum in blue color highlights the last spectrum in which dark excitons can be resolved before being inglobated in the strong emission from the bright exciton complexes.”

Here is the response to the second question about the potential laser-induced heating in our experiments. In the experiments, we used a CW laser at a wavelength of 532 nm with a chopper frequency of 500 Hz and a duty cycle of 25%. We also kept the laser power density much below the damage threshold of the sample.

We add the following brief description of our laser excitation protocol to the SI to provide more clarity on this point:

“We perform the experiments with a CW laser at a wavelength of 532 nm with a chopper frequency of 500 Hz and a duty cycle of 25%.”

In the model of dark exciton interactions, the authors assume that dark excitons have a permanent out of plane dipole moment of $d \sim 0.5$ nm that results in dipole-dipole interactions V_{pp} . While the optical transition dipole of the dark exciton transition is indeed in the z direction, this does not necessarily imply that these excitons have a permanent out-of-plane electric dipole moment, and that it is as large as 0.5 nm.

Early in the text, the authors state: “[...] their permanent out-of-plane dipole - analogous to the one of interlayer excitons in van der Waal heterostructures - enables the control of their spectral properties by an external electric field¹⁴” referencing Zhou et al Nat. Nano (2017). Zhou et al show an out of plane optical transition dipole, but do not show an out of plane electric dipole: there is no evidence of a Stark shift of the dark exciton peak due to an applied electric field. The $d \sim 0.5\text{nm}$, which is larger than the thickness of a single layer TMD, is obtained from Reference 23, a theory paper by Li et al, PRB (2022), which calculates the size of the electron and hole wavefunctions in the z direction, which to lowest order have a gaussian shape of size 0.5nm . This is not the electron hole separation in the z direction in the simplified dipole picture. The authors should demonstrate that dark excitons have an out of plane permanent electric dipole moment, either with original experimental data, or by citing previous experimental works. In addition, they should provide more evidence that $d \sim 0.5\text{nm}$. If not, the authors should not include this dipole-dipole term in the analysis of dark exciton interactions, since in any case the exchange energy contribution is the dominant one. Can the authors also provide a more quantitative comparison of the exciton-exciton interaction strength of dark and bright excitons, either from their data or comparing to previous literature? How does this estimate also compare with the data obtained from the power dependence of the spectra in figure 3? What is the diffusion constant if they use the dark exciton lifetimes provided in the literature?

We thank the Reviewer for raising this important point regarding the theoretical understanding of our experimental results. Prompted by the Reviewer’s comment we revise our model and follow her/his suggestion to consider only exchange interaction which is indeed the dominant term. A detailed discussion on this new model and the list of modifications to the new version of the manuscript are in the initial general remarks “RECONSIDERATION OF DIPOLE-DIPOLE INTERACTION” and “NEW CALCULATIONS OF THE EXCITON DENSITY”. Here we just summarize the main conclusions.

We agree with the Reviewer that the assumption of a permanent dipole moment is not supported by experimental evidence. Moreover, as pointed out by the Reviewer, the exchange interaction term is anyways predominant. Therefore, we follow the reviewer's suggestion and update our dark exciton interaction model to consider Coulombic interactions without dipole-dipole interactions, as proposed in previous literature (Ref.42 and Ref.43 in the main text). We have found that the exchange contribution is dominant in both cases, consistent with the reviewer's observation.

Regarding the quantitative comparison of the Coulomb exchange interaction strength between dark and bright excitons, we have theoretically estimated the densities of bright and dark excitons. In fact, since exciton interaction is dominated by the exchange term $U_{ex} \sim a_0^2 E_b$, its strength is expected to be similar for bright and dark excitons. Our analysis concludes that the dark exciton interaction energy ($\Delta E = n_0 U_{ex}$) is larger than the one of bright exciton due to its much higher density compared to bright excitons. We used a simple theoretical model of

coupled rate equations, which includes both radiative and non-radiative decay pathways, to estimate the density of bright and dark excitons in our system. Despite the approximation of the model, we find good agreement between the calculated and measured density of dark excitons. We refer to the general remarks at the beginning of our response for all the details regarding the model, the results of the calculations and the comparison with the experimental data.

Regarding the comparison of exciton-exciton interaction with power-dependence spectra, we would like to note that it is challenging to exactly quantify exciton-exciton interaction from the power-dependent spectra because bandgap renormalization and Coulomb screening contribute to a redshift in the spectra, particularly at the excitation center. As discussed in the previous comment, at high power, the spectrum is dominated by charged biexcitons that make it impossible to precisely extract the emission energy of dark excitons as a function of pump power. Nonetheless, as mentioned in the manuscript, the experimental observation of the expansion of the exciton cloud with increasing laser excitation power, and the uphill transport of excitons, provide convincing evidence of the strong interaction between dark excitons.

In addition, the reviewer has suggested estimating the diffusion constant. In response to this, we use the approach of L.A. Jauregui et al. (Science 366, 870 (2019)) to estimate the diffusion length and diffusion constant using the dark exciton lifetimes reported in literature. We include this additional analysis in the revised manuscript to provide a more complete understanding of our results. For the purposes of estimation, we use a lifetime of 250 ps and we estimate the diffusion constant according to $D = L_D^2 / \tau$. We find that the diffusion constant (D) ranges from 68 to 240 cm²/s in the unstrained flat energy direction (+x), while it ranges from 40 to 110 cm²/s in the strained uphill energy direction (-x). The diffusion length increases with excitation power caused by repulsive interaction among excitons. We would like to note that Cadiz et al. (Appl. Phys. Lett. 112, 152106 (2018)) reported a diffusion constant for dark excitons in WSe₂ of ~250 cm²/s in agreement with our results. We refer the Reviewer to the general remark "MEASUREMENT OF THE DIFFUSION LENGTH" for all the details about this calculation and the list of changes to the manuscript.

Finally, some additional minor comments:

-The authors state at the beginning of the manuscript that: "[...] bright excitons in which the electron-hole Coulomb bound pair is formed by an electron and a hole with parallel spin at the K (or K') valley", and that dark excitons have electrons and holes with antiparallel spin directions. While it is certainly true that the bright transition is between electron bands with the same spin orientation and the dark transition is between bands with opposite spin orientation, the above statement might cause some confusion given the convention of holes having the opposite spin as the electron band they are in (see for example: Yu et al Arxiv 1507.08103 or Tang et al Nature Comm 10, 4047 (2019))

The Reviewer is correct and this statement as reported in the old version of the manuscript might cause confusion. While it is true that holes have the opposite spin as the electron band

they are in, the statement as written suggested that the spins of the electron and hole in the Coulomb bound pair are aligned, which is not the case.

To clarify, bright excitons are formed by the Coulomb interaction between an electron and a hole in the same spin-valley state at the K (or K') valley. Dark excitons are formed by the Coulomb interaction between an electron and a hole in different spin-valley states at the K (or K') valley. The bright transition is between electron bands with the same spin orientation, while the dark transition is between bands with opposite spin orientation.

We correct this in the introduction of the manuscript. The new sentences read: “These aspects have been mostly explored in bright excitons, which are bound states formed by an electron and a hole in electron bands with the same spin orientation at the K (or K') valley.” and “Spin-forbidden excitons are made of carriers in electron bands with opposite spin orientation in the same valley (K and K') and, contrarily to their bright counterpart, they possess an out-of-plane transition dipole and two orders of magnitude longer lifetime.⁷⁻¹⁰”

-The spectrum in figure 1 has many peaks, it would be useful to include a zoomed in version in the supplementary where the main peaks (dark exciton, dark trion, intervalley dark exciton, and some phonon replicas are highlighted). How do the authors select the energy of these peaks? If a fitting is performed, it should be included.

We thank the Reviewer for this suggestion. In the new version of the manuscript we add a new Figure in the Supplementary material (Figure S4) in which we show a zoom of the most relevant dark exciton peaks, and an example of multipeak fitting used to extract the energy of the peaks. To extract the peak energy, we fit our data with Gaussian functions. In the case of D^0 and I^0 where the weak peaks are very close to the stronger bright trions, we use a multi-peak fitting procedure as shown in Figure S4. The lower energy peaks, such T_1 and D_p^- are well spaced for low pump power and they can be nicely fitted with single Gaussian functions. For the sake of clarity, the new Figure S4 is shown below. Note that, prompted by the next question from the Reviewer, we performed new polarization dependent measurements that are now in Fig.1 of the main text. The old spectra are now in Figure S4a.

This new part is in Note 3 of the SI and reads:

“ In Figure S4a we show the emission spectra of WS₂ at T = 7K taken at a different position with respect to the one shown in Fig.1b of the main text. The identification of the peaks is consistent across the sample. Fig. S4b illustrates the multipeak fitting procedure used to deconvolute overlapping peaks and to extract their energy. Figure S4c is a zoom in the spectra range of the dark trions and phonon replicas to better visualize their peaks. ”

Figure S4: Supplementary data and analysis for peak identification. **a** - Emission spectra of WS₂ at T = 7K excited with σ^+ circular polarization and collected with σ^+ (blue line) and σ^- (red line). Top panel shows the chirality of the emission $\rho = \frac{I^+ - I^-}{I^+ + I^-}$, where I^+ and I^- is the emission intensity of σ^+ and σ^- polarized light, respectively. The peaks of bright and dark exciton complexes are highlighted by vertical dashed lines. **b** - Result of the Gaussian multi-peak fitting in the region of the spectrum of D^0 and I^0 . **c** - Zoom of the spectral range that includes dark triions and phonon replicas.

-The authors should explain why D^0 seems to be showing very strong chiral emission, given that it should be largely suppressed (for example in He et al Nature Comm 11, 618 (2020)).

We thank the Reviewer for this comment that brings to our attention an important point regarding the chirality of neutral dark excitons. We agree that the chirality of spin-dark excitons and spin-dark triions should be suppressed. To clarify this point, we perform new polarization-dependent measurements and try to achieve a better extinction ratio in our optical setup. However, the new measurements confirm the previous observation. In our new experiments, we measure the degree of polarization for the spin-dark triions D^- to be $\sim 1\%$, as it is expected. This result indicates that our measurements are reliable and consistent with previous observations. We attribute the high degree of polarization observed for the natural spin-dark exciton D^0 to its very small spectral weight. In fact, the D^0 peak is surrounded by two

much stronger emissions, the X_s^- above and an unknown peak around 3 meV below. The latter is strongly polarized and, despite being observed in several works (i.e. He et al Nature Comm 11, 618 (2020)), its origin is still unknown. We note the investigation of the origin of this peak is out of the scope of this work, and its presence does not affect the conclusions of our work.

To confirm that the high degree of polarization of D^0 is an artifact, we perform a multi peak fit analysis on the spectral range of D^0 . The results, shown below, indicate a similar intensity of D^0 for the co- and cross-polarized spectra, with a degree of polarization of $\sim 7\%$, much smaller than the one obtained by calculating the degree of polarization from the full spectrum. Moreover, we find a polarization degree for the unknown peak (labeled with “?” in the figure) around 35%, confirming its strong polarization.

To account for this new analysis we make a few modifications to the manuscript. In Fig.1b we have plotted the new polarization-dependent spectra and moved the old ones to the SI in Figure S4. In the main text, when discussing the Fig.1b, we add the following sentence: “We note that the chirality of D^0 should be suppressed due to its out-of-plane optical transition.²³ The nonvanishing chirality of D^0 that emerges in our measurements is due to the nearby presence of the strong emission of X_s^- and of a strongly-polarized unknown peak at ~ 3 meV below. A more careful analysis returns a much lower degree of polarization for D^0 of 7% (see Note 3 of the SI).”

In the SI, we add the results of the multipeak fitting as Figure S5, and the corrected values of the degree of polarization for D^0 . The new Figure S5 is below.

Figure S5: Degree of polarization of D^0 . The deconvoluted peaks are shown for the emission spectra of WS_2 at $T = 7\text{K}$ (shown in Figure 1 of the main text) excited with σ^+ circular polarization and collected with σ^+ (left panel) and σ^- (right panel). The leftmost peak is labeled with “?” as its origin is unknown. The degree of polarization $\rho = \frac{I^+ - I^-}{I^+ + I^-}$ for the deconvoluted peak of D^0 is around 7%.

-What laser power was used in figure 2b-e? Was it CW or chopped? This should be included more clearly in the text or caption.

For the experiment of Figure 2c-d we use a CW laser with power of 600 μW . We include this information in the caption of the figure.

-Can the authors explain more clearly why in figure 4 and S8 some of the bright exciton peaks are just as prominent away from the excitation spot as on the excitation spot? Can lower powers or a more focused collection path reduce this issue?

There are two main reasons for observing strong emission from the bright exciton complexes far away from the exciton spot in Fig.4b,c. On one hand, the excitation point exhibits the strongest emission, and the bright peaks have an in-plane dipole that results in out-of-plane emission that can couple with our detection system even a few micrometers away from excitation. In fact, the emission from these bright peaks can be detected at several micrometers of distance, making it difficult to avoid their collection even with a high-resolution confocal setup. On the other hand, each spectrum in Figure 4, namely each column of the plots, is normalized to its maximum to better visualize the weak emission from dark exciton species. In the waterfall plot of Figure S12, one can clearly see that the bright negative biexciton is still dominating the spectrum at 6 μm away from the excitation spot even in the uphill case. This is the reason why the emission intensity of bright peaks seems almost constant in Figure 4. Without normalization of the spectra, the emission intensity of bright biexcitons decreases by four orders of magnitude as a function of the distance, as shown in Figure S13 below. Moreover, we should stress that the detection and spatial filtering efficiency of our experimental apparatus decreases when detecting with the second galvanometer far away from the excitation spot. This is because in our experimental scheme, we decouple excitation and detection with galvanometer mirrors and the detection is done at larger angles which decrease the coupling with the objective lens.

Figure S13: **a** - Acquisition time as a function of distance used in the experiments with two galvanometers. The time required for acquisition increases by three orders of magnitude as the distance from the center of the exciton increases in order to acquire PL spectra. **b** - PL counts per second of bright exciton and charge biexcitons as a function of distance as extracted from the raw data of Fig. 4c.

In response to the Reviewer's suggestion of using weaker power, we would like to note that this experiment is extremely challenging as it requires the right balance between the interaction energy of dark excitons (that is power dependent), emission intensity of all the exciton species (that is also power dependent) and the height of the energy landscape. In particular, for lower powers, the energy shift of dark exciton peaks decreases as well as their emission intensity, making harder to resolve their peaks in contrast to the one of trions or biexcitons. For stronger powers, the emission of bright excitons becomes even more predominant. We found that an excitation power of 150 μW in the energy landscape of Fig. 4, gave the best result in terms of peak separation and contrast, allowing us to clearly observe the propagation of dark excitons in the uphill experiments. **However, we performed similar experiments by varying the pump power and the energy landscape. These new measurements are now presented as Figure S15-17 in the SI.**

First, we note that in Fig. 4 we use a pump power of 150 μW and dark excitons already propagate to the top of the energy ramp of height 20 meV. Therefore, increasing the power would not add more information as the dynamics is limited by the energy landscape. In Figure S15 we compare the uphill diffusion in the same position of the sample with a pump power of 150 μW and 100 μW . In the latter case, the emission of dark excitons vanishes beyond a distance of $\sim 4.5 \mu\text{m}$, which corresponds approximately to 12 meV uphill diffusion. This behavior seems to indicate that uphill transport is now limited because the blue shift induced by dark exciton interaction cannot fully overcome the energy barrier. However, we cannot exclude that the signal from dark excitons gets too weak to be detected. Conducting measurements at lower power is challenging because the signal from diffused excitons decreases exponentially away from the excitation center. In Figure S16, we present results for uphill diffusion measurements

performed on a smaller energy landscape with a height of 4 meV that turns into a region with a “flat” energy landscape. These measurements are taken at a laser power of 100 μW . Figure S16b shows that bright excitons, trions, and biexciton complexes do not move uphill, consistent with our previous observations. However, dark excitons are able to overcome the energy barrier and reach the plateau at the end of the energy ramp. Although we observe strong emissions from dark excitons beyond the uphill region, the emission from the plateau does not provide any further information about the actual blueshift.

Figure S17 shows the diffusion across a flat and inhomogeneous region. In this experiment, D^0 and I^0 emerge from the bright trion shoulders only when the energy landscape changes by a few meV.

As a final note, we would like to stress that the conclusions on dark exciton propagation due to interaction is supported by the energy shift of the emission peaks in the uphill energy landscape, and the intensity of the dark excitons does not provide further insight.

Reviewer #2 (Remarks to the Author):

The authors present a (density-dependent) study of energy as well as space-resolved time-integrated signal in transition metal dichalcogenides monolayers. In particular they find three different behaviours between spin-dark and bright states: i) only the bright states show a redshift in energy with increasing density and ii) only the dark states show an increased diffusion length with increasing density iii) only the dark excitons are able to diffuse uphill. The authors attribute these effects to a vertical spatial separation between electron and hole of 0.5 nm for the only dark states. While the results are interesting, the conclusions for the moment are not enough supported by the evidence and in general the analysis shows still too many flaws to deserve publication in Nature Communications.

We thank the Reviewer for the time spent reviewing our manuscript, and for the valuable suggestions. We are happy she/he finds our results interesting. In the following we address all the points raised and solve all the flaws highlighted. We hope he/she will find our explanation and changes satisfactory.

In response to the reviewer's comments, we have made several changes to our analysis. Specifically, we formulate a different model for the exciton-exciton interaction using only exchange terms and removing dipole-dipole interaction. We also include additional data and analysis to support our conclusions to make the presentation of our results more clear and coherent. Since similar concerns were shared by other Reviewers, we add a detailed discussion and description of the new model as a "**General remarks**" at the beginning of this response. We refer the Reviewer to these general remarks for the details on the new interaction model and for a list of the major changes to the paper.

We hope that the Reviewer will find our improvements satisfactory and we thank her/him again for her/his time and valuable feedback.

In particular the authors should

- strongly discuss the relationship between electron-hole spatial separation and the so-called optical dipole moment, in particular presenting literature clearly connecting these two quantities for the case of spin-dark excitons in monolayers. In addition comparison to the case of vertical heterostructures should be presented in relation with the polarization displayed by interlayer excitons.

We appreciate the reviewer's suggestion to discuss the relationship between electron-hole spatial separation and the optical dipole moment for dark excitons in TMD monolayers. In our initial analysis, we included dipole-dipole interaction in the model for dark exciton interactions. However, the exchange interaction is always predominant over the dipole term. Therefore, as discussed extensively in the initial general remarks we exclude the dipole term from our analysis and consider only the exchange interaction. With this new approach, we show that the stronger interaction observed in dark excitons is due to their higher density when compared to bright

excitons. We refer the Reviewer to the general remarks RECONSIDERATION OF DIPOLE-DIPOLE INTERACTION” and “NEW CALCULATIONS OF THE EXCITON DENSITY” for a detailed description of the new model.

Regarding the comparison to vertical heterostructures, which have interlayer excitons, we agree that the polarization displayed by interlayer excitons is strongly dependent on the separation between the electron and hole wave functions in the vertical direction, leading to a permanent dipole and dipole-dipole interaction. However, in our study, we model the interaction between dark excitons as exchange interactions, without considering dipole-dipole terms. Therefore, we believe that the comparison of dark exciton to interlayer excitons may no more directly relevant for this work. However, in the manuscript we add comparison on the diffusion properties of dark and interlayer excitons, which are relevant to the focus of this work. We appreciate the Reviewer’s comment that has triggered significant improvements to the analysis of our results.

- The authors state that the absence of redshift for dark states is due to the presence of a blueshift canceling the redshift (present for bright states). While it is clear to me that spatially-separated electron-hole would present such a blue-shift in the presence of electric field, it is not equally clear that this has already been observed for the spin-dark states in monolayers. The authors should clearly present literature about this.”

The reviewer’s question regards the power-dependent spectra that we presented in Fig.3 of the previous version of our manuscript. In particular the Reviewers ask for clarification regarding the observation of the blueshift in dark excitons. We would like to clarify that in our study, the repulsive interaction among excitons, rather than an external electric field, is responsible for the observed blueshift. In the power-dependent measurement of the spectral emission, this blueshift partially cancels out the redshift due to bandgap renormalization. Here, we assumed that the bandgap renormalization effects and Coulomb screening are the same for the bright and dark excitons because they originate from the same material and are subject to the same dielectric environmental conditions. Previous studies have shown that these effects are typically material-dependent [Nat Comm. 8, 15251 (2017), ACS Nano 11, 12601 (2017), Nano Lett. 14, 3743(2014)]. However, we realize that this is just an assumption and, to the best of our knowledge, there are no reports of dedicated measurements on the effect of band gap renormalization on specific excitonic complexes. After a careful analysis and prompted by the questions from other reviewers, we tone down the discussion on the power-dependent exciton blueshift and move the power dependent spectra to the SI. The reason for this choice is that this piece of data, despite supporting our conclusion, does not allow for a quantitative analysis and does not add much to the discussion on the interaction-driven enhanced transport of dark excitons, which is the main focus of our work. We would like to stress that the increase of the diffusion length as a function of the pump power, and the uphill diffusion can be quantitatively analyzed and represent clear and more relevant evidence of interaction-driven transport of dark excitons.

The quantitative analysis of the power-dependent spectra is limited by two factors. On one hand, it is challenging to precisely extract the energy shift of dark excitons because the spectrum at high power is fully dominated by charged biexcitons and a precise peak deconvolution is impossible. Second, without precisely knowing how bandgap renormalization affects different exciton complexes we cannot quantitatively compare the energy shift of individual excitons due to interactions. However, we observed a different shift rate for bright and dark exciton species which indicate a different interaction. This shift can be quantified only up to medium laser powers (corresponding to the blue spectrum in Fig. S7). Therefore, we move these data to the SI as Figure S7 and better highlight the different energy shifts.

In the main text, we add the following discussion to clarify the effects and Coulomb screening in power dependent spectra:

“The evolution of the emission spectrum for increasing pump power, reported in Fig. S7 in the SI, shows a smaller redshift due to band renormalization for dark exciton species when compared to the bright counterparts.^{32–34} This suggests the dark excitons are affected by stronger repulsive interactions.”

The second part of the question regards Stark shift in dark excitons. Although Stark shift in spatially-separated electron-hole pairs with permanent dipole is well-established in the literature, to the best of our knowledge there is currently no literature demonstrating such an effect in spin-dark excitons in monolayers.

We appreciate the valuable input from the Reviewer and we take it into account in the revised manuscript. In response to this reviewer’s comment, we revise our model and discussion in the manuscript. This new approach provides a more accurate description of the system and helps to better explain our experimental results. Again, we refer the Reviewer to the general remarks “RECONSIDERATION OF DIPOLE-DIPOLE INTERACTION” and “NEW CALCULATIONS OF THE EXCITON DENSITY” for a detailed explanation.

- Concerning the density-dependent increase of the diffusion (length) shown for example in Fig. 3f: Can the authors distinguish between an increased diffusion coefficient and/or an increased recombination time?

The Reviewer has raised an interesting point about the observed density-dependent increase in the diffusion shown in Figure 3. Specifically, the Reviewer asks whether the increase is due to an increased diffusion coefficient, an increased recombination time, or both.

To address this question, we conduct an additional analysis to better quantify the increase in the diffusion by estimating the diffusion length as a function of the pump power.

We follow the approach used by L.A. Jauregui et al. (Science 366, 870 (2019)) in a previous study. This information is now included in the revised manuscript in Fig.3f to provide a more comprehensive presentation of our results. The details and the results of this new analysis are discussed in the general remark “MEASUREMENT OF THE DIFFUSION LENGTH” at the beginning of this response.

Here, we would like to note that we find that the diffusion length for dark excitons increases as a function of power, while the one of bright excitons stay constant. This indicates the presence of additional interaction energy in the population of dark excitons. Moreover, we estimate the diffusion constant according to $D = L_D^2/\tau$ where, L_D is the diffusion length and τ the radiative lifetime. By using a constant value for the lifetime of dark excitons as reported in the literature we find an increasing diffusion constant for dark excitons from 68 to 240 cm² as a function of the pump power. We note that these values are in agreement with the previous measurements on WSe₂ reported by Cadiz et al. Appl. Phys. Lett. 112, 152106 (2018).

To clarify this point in the manuscript when discussing exciton diffusion, we add a sentence regarding the diffusion length: “we attribute the increase of diffusion length of dark excitons to the large interaction energy.”

Although we do not have the experimental capabilities to perform time-dependent measurements of dark exciton diffusion because our temporal resolution is limited to ~350 ps which is slower than the lifetime of excitons in WS₂, we would exclude the possibility that the recombination time of excitons increase as a function of the exciton density. On the contrary, there are several reports that show how recombination time decreases when the exciton density increases due to the onset of exciton-exciton annihilation. This effect dramatically reduces in hBN-encapsulated monolayers. However, the reduction of radiative recombination at high pump fluence in WS₂ has been reported for example in Zipfel et al., PRB 101, 115430 (2020). A similar effect has been reported for interlayer excitons by L.A. Jauregui et al., Science 366, 875 (2019). Furthermore, our theoretical calculation on the exciton density suggests that exciton annihilation is not significant with the pump power used in the experiments. More details on this can be found in the general remark “NEW CALCULATIONS OF THE EXCITON DENSITY”. Therefore, we can consider a constant radiative lifetime as a function of the pump power. We add Zipfel et al., PRB 101, 115430 (2020) as reference in the SI.

- Where does the value of 0.5 nm as a dipole come from? It seems very similar to the values found for interlayer excitons in heterostructures; however I would have expected a much smaller value for monolayer excitons.

We thank the Reviewer for this question regarding the value of 0.5 nm indicated as the dipole size in the previous version of the manuscript. However, in our revised manuscript, we remodel the interaction among dark excitons as exchange interactions rather than dipole-dipole interaction. Therefore, the concept of a permanent dipole and dipole length are no longer used in our analysis to understand our experimental results. We refer the Reviewer to the initial general remarks for more details about the new model. To answer the Reviewer’s question, the value of 0.5 nm was based on the theoretical work by Li et al. (PRB 106, 085414 (2022)), which approximated the out-of-plane confinement of monolayer excitons to a one-dimensional quadratic quantum well. The ground state of this well is Gaussian in shape and can be characterized by a parameter “ a_z ,” which has a value comparable to twice the thickness of the

monolayer. We hope this explanation helps to clarify the origin of the dipole length used in our previous analysis.

- The density-dependent blueshift as well as increase of the diffusion length should be quantitatively compared to those found in literature for dipolar interlayer excitons.

We thank the Reviewer for this question that has triggered significant improvements in the analysis of our experimental results and the calculation of the diffusion length. In the initial general remark "MEASUREMENT OF THE DIFFUSION LENGTH" we discuss the details of the estimation of the diffusion length of bright and dark excitons in different energy landscapes.

Here, we compare our results to the ones from interlayer excitons in a heterostructure reported by Sun et al. (Nature Photonics 16, 79 (2022)) and by Jauregui et al. (Science 366, 875 (2019)). L.A. Jauregui et al. found a max blueshift of ~10 meV and an increasing diffusion constant from ~1 to ~2 μm with increasing excitation power of three orders of magnitude. Sun et al. reported a blueshift of ~20 meV, a propagation distance of ~1.8 μm and an estimated interlayer exciton density of $\sim 10^{11} \text{ cm}^{-2}$. In the case of dark excitons, we observe a blueshift of 20 meV (limited by the strain landscape), an increasing diffusion length from 1.4 to 2.4 μm with two and a half orders of magnitude increase of the laser excitation power. The diffusion length of dark excitons is slightly longer than the one of interlayer excitons. We note that our value is in agreement with the one measured by Cadiz et al. (Appl. Phys. Lett. 112, 152106 (2018)) that measured a diffusion length of 1.5 μm for dark excitons in WSe₂. The comparable diffusion length between dark and interlayer excitons despite the shorter lifetime could be explained by the different interaction. Dark excitons show slightly larger blue-shift which can be attributed to their stronger binding energy. However, as already discussed in one of the previous comments, we cannot extract quantitative information regarding the blue-shift from the power-dependent spectra and a more direct comparison between the blueshift would be misleading. Nevertheless, our measurements on the power-dependent diffusion and the good match between the calculated and measured dark exciton density allow us to identify exchange interactions as the predominant interactions for the long-range diffusion.

We follow the Reviewer's suggestion and compare the diffusion of dark and interlayer excitons. We add the following sentence:

"Similarly to interlayer excitons,^{16,17} exciton interactions promote longer diffusion. We note how the diffusion length of dark excitons is comparable to the one of interlayer excitons despite the shorter lifetime (τ), suggesting a different interaction strength."

- When discussing the uphill diffusion, the authors write "clearly indicates a large repulsion energy that allows them to overcome the potential gradient". This statement should be supported by density-dependent studies.

We appreciate the reviewer's suggestion to improve our understanding of uphill diffusion. We would like to note that this experiment is extremely challenging as it requires the right balance

between the interaction energy of dark excitons (that is power dependent), emission intensity of all the exciton species (that is also power dependent) and the height of the energy landscape. In particular, for lower powers, the energy shift of dark exciton peaks decreases as well as their emission intensity, making harder to resolve their peaks in contrast to the one of trions or biexcitons. For stronger powers, the emission of bright excitons becomes even more predominant. We found that an excitation power of 150 μW in the energy landscape of Fig. 4, gave the best result in terms of peak separation and contrast, allowing us to clearly observe the propagation of dark excitons in the uphill experiments. **However, we performed similar experiments by varying the pump power and the energy landscape. These new measurements are now presented as Figure S15-17 in the SI.**

First, we note that in Fig. 4 we use a pump power of 150 μW and dark excitons already propagate to the top of the energy ramp of height 20 meV. Therefore, increasing the power would not add more information as the dynamics is limited by the energy landscape. In Figure S15 we compare the uphill diffusion in the same position of the sample with a pump power of 150 μW and 100 μW . In the latter case, the emission of dark excitons vanishes beyond a distance of $\sim 4.5 \mu\text{m}$, which corresponds approximately to 12 meV uphill diffusion. This behavior seems to indicate that uphill transport is now limited because the blue shift induced by dark exciton interaction cannot fully overcome the energy barrier. However, we cannot exclude that the signal from dark excitons gets too weak to be detected. Conducting measurements at lower power is challenging because the signal from diffused excitons decreases exponentially away from the excitation center. In Figure S16, we present results for uphill diffusion measurements performed on a smaller energy landscape with a height of 4 meV that turns into a region with a “flat” energy landscape. These measurements are taken at a laser power of 100 μW . Figure S16b shows that bright excitons, trions, and biexciton complexes do not move uphill, consistent with our previous observations. However, dark excitons are able to overcome the energy barrier and reach the plateau at the end of the energy ramp. Although we observe strong emissions from dark excitons beyond the uphill region, the emission from the plateau does not provide any further information about the actual blueshift.

Figure S17 shows the diffusion across a flat and inhomogeneous region. In this experiment, D^0 and I^0 emerge from the bright trion shoulders only when the energy landscape changes by a few meV.

As a final note, we would like to stress that the conclusions on dark exciton propagation due to interaction is supported by the energy shift of the emission peaks in the uphill energy landscape, and the intensity of the dark excitons does not provide further insight.

Finally, we would like to stress that the statement on large repulsion energy is already supported by two pieces of evidence. The first is the expansion of the dark exciton cloud and the increase of their diffusion length as a function of the excitation power, namely as a function of the dark exciton density. The longer diffusion is explained with an increase of the interaction energy due to the larger density. The second is the uphill propagation. Since dark excitons are generated at the bottom of the energy ramp and observed to recombine on top of the ramp 20 meV above and 6 μm away, they must be repelled from the excitation spot with an extra energy of at least

20 meV. This repulsion is explained by interaction energy. We believe that these experimental studies are enough to support the presence of repulsion and interaction energy.

We modify the sentence pointed out by the Reviewer to include the considerations above. The new sentence read: “Surprisingly, dark exciton complexes show a remarkable blueshift as a function of the distance, indicating that they overcome the potential gradient. Dark excitons can diffuse away from the excitation spot, relax to the energy landscape - which represents the lowest energy state available to them at that particular position - and then recombine radiatively. This observation together with the increasing of the diffusion length as a function of pump power indicate the buildup of interaction energy in the dark exciton population.”

In the SI, we add we add the new data of uphill diffusion with different powers and different energy landscapes as Figure S15, S16 and S17, together with the following discussion:

“In Fig. S15a, the excitation power is 150 μW , while in Fig. S15b is 100 μW . Dark excitons can diffuse across the full uphill energy landscape of 20 meV in the former case. However, in the latter case, the emission of dark excitons vanishes beyond a distance of $\sim 4.5 \mu\text{m}$, which corresponds approximately to 12 meV uphill diffusion. This behavior seems to indicate that uphill transport is now limited because the blue shift induced by dark exciton interaction cannot fully overcome the energy barrier. However, we cannot exclude that the signal from dark excitons gets too weak to be detected. Conducting measurements at lower power is challenging because the signal from diffused excitons decreases exponentially away from the excitation center.

In Figure S16, we present results for uphill diffusion measurements performed on a smaller energy landscape with a height of 4 meV that turns into a region with a “flat” energy landscape. These measurements are taken at a laser power of 100 μW . Figure S16b shows that bright excitons, trions, and biexciton complexes do not move uphill, consistent with our previous observations. However, dark excitons are able to overcome the energy energy barrier and reach the plateau at the end of the energy ramp. Although we observe strong emissions from dark excitons beyond the uphill region, the emission from the plateau does not provide any further information about the actual blueshift.

Figure S17 shows the diffusion across a flat and inhomogeneous region. In this experiment, D^0 and I^0 emerge from the bright trion shoulders only when the energy landscape changes by a few meV.”

Reviewer #3 (Remarks to the Author):

The authors report on interaction driven transport of dark excitons at the K point of a WS₂ monolayer at low temperatures (7K).

In a first step they identify the different excitonic features in their PL spectra by comparing with literature, identifying dark and bright excitons. Secondly, the authors investigated the diffusion behavior of the different exciton species. While the bright excitons show low diffusion, the dark excitons spread over the whole sample area. The authors attribute the large diffusion of dark excitons to their long lifetime and exciton-exciton interaction induced by the permanent dipole which generates a drift potential. To demonstrate the strong interaction of the dark excitons power dependent diffusion and spectral measurements were performed. While the bright excitons show a redshift due to band normalization and a reduction of the effective bandgap with increasing power, the dark excitons show no/less redshift, indicating a blueshift generated by the exciton-exciton interaction. In addition, the diffusion spot increases for the dark excitons with increasing laser power.

After this the authors turn to the main claim of the paper the investigation of exciton diffusion in a varying energy landscape.

1) excitation and emission on the same spot: All excitons show a redshift of 20meV due to the increasing strain potential over 6 micrometer, leading to an maximum strain of 0.4%

2) excitation in the low strain region and detection for increasing strain values (energy downhill): The bright neutral excitons shows a redshift due to the funneling effect, same holds for the dark excitons where in addition to the funneling also repulsion interaction plays a role.

3) excitation at high strain and detection at various lower strain values (energy uphill): all bright excitons show no energy shift. In contrast the dark excitons show a strong blueshift indicating that they diffuse uphill and recombine at a position with lower strain (higher energy). The authors conclude that the high repulsion energy allows the dark excitons to overcome the potential gradient. They claim that the different diffusion behavior is due to the weak interaction of the in-plane dipoles of the bright excitons compared to the strong interaction of dark excitons with out-of plane dipoles.

From the uphill diffusion the authors estimate a repulsive interaction of 20meV from which they calculated a dark exciton density of $2.5 \times 10^{11} \text{ cm}^{-2}$.

In my opinion the paper is reporting on an interesting finding, however the explanation is in the current form not very convincing to me. In addition the argumentation is somehow confusing for me.

Therefore, I do not recommend publication in Nature Communications in the current form.

We thank the Reviewer for carefully reviewing our manuscript on dark exciton interactions and transport in transition metal dichalcogenides monolayers. We also appreciate the reviewer's positive assessment of our finding as interesting.

We understand that the explanation and argumentation in the old form was not entirely convincing to the reviewer. In this new version of the paper, we address the Reviewer's comments with great care and make several revisions to improve the clarity and coherence of our presentation. In particular, we remodel the exciton-exciton interaction using only exchange. We also provide additional data and analysis to support our conclusions. We would like to note that significant changes have been made to the manuscript and they are presented in detail at the beginning of the response as "**General remarks**". They include the new model to understand dark exciton interactions, new theoretical calculations of the dark exciton density, and the measurement of the diffusion length for dark and bright excitons.

We carefully review the detailed comments provided by the Reviewer and we address each of them in the revised manuscript. We provide more detailed explanations and include additional references to support our arguments. We hope that these revisions will make our findings and conclusions more compelling to the Reviewer and other readers.

Overall, we believe that revised manuscript presents significant improvements and we hope that the Reviewer will find it suitable for publication in Nature Communications. We thank the reviewer again for the valuable feedback.

In the following I have several detailed comments where in my opinion the study is not convincing:

- In Fig. 2 the authors show the diffusion spots of dark and bright excitons. For the dark excitons there seems to be even emission visible next to the flake. Why is that the case?

The white outlines of the monolayer look very different to the image in a. What is the reason?

There are two possible reasons that one can still see some weak dark exciton emission outside the flake region.

- Dark excitons have an out-of-plane transition dipole with in-plane light emission. Therefore, light emitted from dark excitons travels in-plane (this was shown for example by Wang, G. et al. in Phys. Rev. Lett. 119, 047401 (2017)) and can be scattered by flake edges and by interfaces of our hBN/WS₂/hBN heterostructure. An image of the heterostructure used in the experiments is shown in Figure S3. Light scattered from flaked edges is not uncommon for dark excitons and has been reported for instance in Tang et al., Nature Communications 10, 4047 (2019).
- Due to the diffraction limit of our setup, the PL emission from an infinitely small spot will spread to a finite size on the EMCCD camera. This physical limitation of an imaging system makes sharp edges dimmer. This effect has been observed in other similar measurement systems, for instance by L.A. Jauregui et al. Science 366, 870 (2019), and by Z. Sun et al., Nature Photonics 16, 79 (2022).

In Fig. 2 c,d,e and Fig.3a,b,c we draw the white lines based on the images of the sample taken with white light using the same detection path. It is possible that there is a small mismatch

between them due to the different light source used. Fig.2a is a PL map of the sample taken with a different detection path compared to the measurements of exciton propagation (Fig. 2b,c,d and Fig.3a,b,c). Fig.2a is taken in a confocal mode using the galvanometer system and spatial filtering system, as shown in Fig. S1a. The resolution achieved with this optical method (~350 nm) is much better than the one for real space imaging with the EMCCD real space imaging (~1 μm). The different detection method explains the difference in the appearance of Fig.2a compared to the other images. We take care to ensure that the white outlines in Fig. 2 and 3 represent the position of the monolayer flake. In the revised manuscript, we redo the outlines, indicate the flake edges in Fig.2a, and provide additional details on the experimental conditions for each image in Fig. 2.

- The diffusion length of the dark excitons (Fig 2 d, e and Fig 3 d) seems to be limited by the flake edges. Have the authors measured on a larger flake, and can they estimate the diffusion length for the different species?

Prompted by this Reviewer's question, we extract and compare the diffusion length of bright and dark excitons. For all the details of the results and the changes in the manuscript, we refer the Reviewer to the general remark "MEASUREMENT OF THE DIFFUSION LENGTH" at the beginning of our response. Here we just summarize the main conclusions relevant for this question.

The diffusion length of dark excitons strongly depends on the excitation power and the energy landscape. The diffusion length of bright excitons has a constant value. The results are reported in Figure R4 below, which is now included in the manuscript as Fig.3f.

We note that we measure a maximum diffusion length of 2.4 μm for dark excitons. While we have not performed measurements in a larger flake, we believe that our flake size of ~20 μm is sufficiently large for this kind of study. Moreover, it is challenging to exfoliate and assemble in heterostructures larger monolayers with the high quality required for this kind of measurements. Furthermore, we note that flakes of smaller or similar size have been used in the past to perform exciton transport studies with comparable diffusion lengths. A couple of examples for interlayer excitons are Sun et al. Nature Photonics 16, 79 (2022) and Jauregui et al. Science 366, 875 (2019).

We thank the Reviewer for this comment that has triggered significant improvements to the paper.

Figure R4: The diffusion length of bright and dark excitons in strained uphill energy and unstrained flat energy direction. The error bars are the error of the fit.

- Fig 3 a. For me it is surprising that the blueshift due to exciton-exciton interaction of the dark excitons exactly compensates for the redshift of the bandgap reduction. Do the authors have an explanation for that? Can the authors show the fitting procedure for the spectra and give an error estimation for their extracted exciton energies?

We appreciate the reviewer's questions regarding the analysis of the interaction energy in our power-dependent measurements. Before answering, we would like to note that there is not an "exact" compensation between the interaction energy and the band gap reduction. What we observe in our power-dependent spectra is that bright exciton species and dark exciton species shift in energy at a different rate when increasing the pump power. We try to highlight this in a new figure (shown also below) in which we highlight the last spectrum where dark exciton replicas can be resolved. Here we observe a redshift for bright excitons of -0.65 ± 0.05 meV, while the shift of D_p^- is -0.23 ± 0.02 meV. An example of the fitting procedure is shown below in Figure R5.

Moreover, in our initial analysis, we assumed that the bandgap renormalization effects and Coulomb screening are the same for the bright and dark excitons because they originate from the same material and are subject to the same dielectric environmental conditions. Previous studies have shown that these effects are typically material-dependent [Nat Com. 8, 15251 (2017), ACS Nano 11, 12601 (2017), Nano Lett. 14, 3743(2014)]. However, we agree with the Reviewer that this is just an assumption and, to the best of our knowledge, there are no reports of dedicated measurements on the effect of band gap renormalization on specific excitonic complexes.

After a careful analysis and prompted by the questions from other Reviewers, we tone down the discussion on the power-dependent exciton blueshift and move the power dependent spectra to

the SI. The reason for this choice is that this piece of data, despite supporting our conclusion, does not allow for a quantitative analysis and does not add much to the discussion on the interaction-driven enhanced transport of dark excitons, which is the main focus of our work. We would like to stress that the increase of the diffusion length as a function of the pump power, and the uphill diffusion can be quantitatively analyzed and represent clear and more relevant evidence of interaction-driven transport of dark excitons.

The quantitative analysis of the power-dependent spectra is limited by two factors. On one hand, it is challenging to precisely extract the energy shift of dark excitons because the spectrum at high power is fully dominated by charged biexcitons and a precise peak deconvolution is impossible. Second, without precisely knowing how bandgap renormalization affects different exciton complexes we cannot quantitatively compare the energy shift of individual excitons due to interactions. However, we observed a different shift rate for bright and dark exciton species which indicate a different interaction. This shift can be quantified only up to medium laser powers (corresponding to the blue spectrum in Fig. R5). Therefore, we move these data to the SI as Figure S7 and better highlight the different energy shifts.

In the main text, we add the following discussion to clarify the effects and Coulomb screening in power dependent spectra:

“The evolution of the emission spectrum for increasing pump power, reported in Fig. S7 in the SI, shows a smaller redshift due to band renormalization for dark exciton species when compared to the bright counterparts.^{32–34} This suggests the dark excitons are affected by stronger repulsive interactions.”

We move the detailed discussion regarding the power-dependent spectra to the Note 4 of the SI, together with the experimental data. It reads:

“4. Power-dependent spectral emission

Figure S7 shows the emission spectra taken with increasing laser power, from 0.05 μW to 500 μW . At high excitation power, the spectra are dominated by the strong emission of bright excitons (XX^- , X^0 and X_T^- , X_S^-), and the precise deconvolution the emission energy of D_p^- and T_1 can not be done. As previously observed in similar samples, at high carrier density a redshift of the exciton complexes occurs because of band renormalization and a reduction of the effective bandgap.^{14–16} We perform the experiments with a CW laser at a wavelength of 532 nm with a chopper frequency of 500 Hz and a duty cycle of 25%. Therefore, we can exclude major effects due to heating of the sample. As far as D_p^- and T_1 are resolvable in the spectra (blue curve in Fig. S7), it is possible to observe that they experience a smaller energy shift compared to the redshift of bright exciton complexes. The redshift of bright excitons 0.65 ± 0.05 meV, while the shift of D_p^- is 0.23 ± 0.02 meV. This difference indicates that, at high density, dark excitons undergo different interaction processes. However, we do not know how band gap renormalization affects the different exciton complexes and a quantitative analysis is impossible at this moment.”

Figure R5: **a** - Emission spectra at different excitation powers ranging from 0.05 to 500 μ W. The emission energy at low power for the neutral excitons X^0 , the negatively charged biexciton XX^- , the dark exciton T_1 and the dark trion phonon replica D_{K3}^- is highlighted with black vertical dashed lines. The spectrum in blue color highlights the last spectrum in which dark excitons can be resolved before being inglobated in the strong emission from the bright exciton complexes. **b-e** show the fitting procedure for the spectra at low and high power.

- The authors put the energetic blue shift of the dark excitons with increasing excitation power directly in line with their larger diffusion. However, this relation is not directly clear to me. Can they explain in more detail and give references supporting this?

We thank the reviewer for this question that resulted in a better presentation of our results and triggered further clarifications on the role of interaction in the longer transport.

When interaction is present in the exciton population, the total energy changes by $\Delta E = n_0 U_{ex}$, where n_0 is the exciton density and U_{ex} the interaction strength. The exciton density is proportional to the excitation power. For repulsive interaction, the interaction energy is positive and results in a blueshift of the emission. At the same time, ΔE provides a potential gradient for excitons that are pushed away from the excitation spot. The higher the interaction energy, the larger is the potential gradient and the diffusion length increases. This effect emerges clearly in dark excitons in which the diffusion length increases as a function of the density, as shown in Fig. 3 of the manuscript. In our measurements, the role of the interaction energy also emerges

in the uphill propagation in which the interaction energy ΔE allows dark excitons to overcome the energy ramp produced by strain.

This relationship between blueshift and larger diffusion with increasing excitation power has been observed in previous studies on interlayer excitons in heterostructures, including Sun et al. Nature Photonics 16, 79 (2022) and Jauregui et al. Science 366, 875 (2019). In these studies, the authors attributed the repulsive interaction between the excitons to their dipole-dipole interactions and observed that the resulting exciton transport was dominated by drift diffusion. In dark excitons the origin of the interaction is different but still of repulsive nature. Therefore, the dynamics are qualitatively similar.

We hope this explanation clarifies the relationship between the blueshift of the dark excitons and their larger diffusion with increasing excitation power. In the new version of the manuscript, we clarify the role of interaction in the propagation properties of dark excitons by adding the new Figure 5c and its discussion. This new section reads:

“The transport and relaxation pathways for the dark exciton D^0 are schematically illustrated in Fig. 5c. The system is initially excited to create dark excitons as shown in Fig.5a. The high density of dark excitons favors the strong exciton-exciton repulsive interaction that increases the overall energy of the dark exciton population at the excitation position. This repulsive interaction at the excitation position provides initial momentum to excitons which diffuse away even towards locations of the sample with higher energy landscape. The transported hot dark excitons can then thermalize toward the bottom of their energy dispersion via scattering with low energy Γ acoustic phonons.²⁸ From this state they can radiatively recombine in the form of dark excitons D^0 , dark trions D^- , phonon replica, T_1 or I^0 .”

- Where the measurements presented in Figures 1-3 done on the same flake as the measurements in Figure 4 (strain landscape)? If so, why is the strain gradient not affecting the diffusion spots in Figures 1-3? I would expect a distorted diffusion spot due to the funneling effect. Especially for the neutral excitons, the authors show that the excitons move downhill but not uphill, therefore one would expect an asymmetric diffusion spot on a sample with a spatial strain gradient.

In case the measurements in Figs. 1-3 were done on a different sample (without strain gradient) the authors could repeat the diffusion measurement on the sample of Fig. 4. Do they see a distorted diffusion?

We thank the reviewer for this question that has triggered further analysis of our results. We confirm that the measurements presented in Figures 1-3 are performed on the same flake as the measurements in Figure 4. The Reviewer is right in expecting an asymmetric diffusion due to the strain landscape. As discussed in detail in the general remark “MEASUREMENT OF THE DIFFUSION LENGTH”, we do observe a different diffusion length for dark excitons, with a reduction of propagation, in the uphill region. In this measurement, the excitation of excitons occurs at about 2 μm from the beginning of the ramp, so this effect is not appreciable in bright excitons whose diffusion length is constant at around 1 μm . We would like to note that, with this

particular experimental scheme, we are able to observe the diffusion of dark excitons in both strain and unstrain case.

This question triggered significant changes in the manuscript. The diffusion length has been extracted for the uphill and flat regions of the sample. This results in the new Fig.3f. In Fig. 3d we include the intensity profile for both propagation directions to highlight the asymmetry. A discussion on the asymmetry is added to the text and reads:

“Figure 3f shows that L_D increases from 1.4 to 2.4 μm in the unstrained (+x) directions and from 1 to 1.6 μm in the strained (-x) region. The asymmetry in the diffusion length in the strained uphill and unstrained directions is attributed to the different energy landscape. Figure S11 in the SI shows the characterization of strain along the horizontal direction x . After a quick interface region, in the negative x direction, strain creates a smooth and linear uphill energy gradient that limits diffusion. In both directions, the increase of the characteristic diffusion length L_D with increasing excitation power indicates an enhanced diffusion at high exciton densities.”

In the SI, we add the new Figure S11 to better illustrate the geometry of our experiments and highlight the different regions of the sample with respect of the excitation spot.

- The authors should provide a strain map of the whole sample (which can be easily done by mapping the bright exciton energy as a function of position). The authors only show the spectra and strain profile along a line (Fig4), but for the diffusion the strain landscape in 2D is important.

It is also not clear where the measurements are performed on the sample. The authors should indicate the positions for excitation and emission on the sample. I assume the authors measure from one edge of the flake to the other? However, the presence of an edge would be important in the discussion of diffusion directions.

We thank the Reviewer for this suggestion. We agree with her/him that knowing the strain landscape is important in diffusion measurements, thus we add to the new version of the manuscript a more complete characterization of the strain in the relevant parts of the samples. As correctly noted by the Reviewer, we do analyze exciton diffusion on a linear profile. The main results of our work are the observation of a power-dependent diffusion length and the propagation in an uphill energy landscape of dark excitons. The measurement of the exciton diffusion length in Fig.3 is done in a line along the $\pm x$ direction, and the measurement of uphill diffusion is also done along the same direction. We modify Fig.2a to include the position of the laser for the diffusion measurements shown in Fig.2 and 3, and to indicate the different strain regions of the sample. Moreover, in Fig.3a-c we now clearly indicate the lines from which we extract the intensity profiles for the measurements of the diffusion length.

In the new Figure S11, we add the emission spectra along the full $\pm x$ direction showing that the left most part of the monolayer (indicated approximately by a white box in Fig.S11) is characterized by a linear gradient of strain that creates an energy gradient increasing toward the left edge. This is the region where we perform the downhill and uphill diffusion experiments of Fig.4. The right most part of the sample (indicated approximately by a yellow box) shows much

smaller variations of strain creating an almost “flat” energy landscape. The two regions are separated by a transition region in which strain changes quickly. In this transition region, the spectral emission is noisy and the linewidth of the excitation peaks increases significantly. In the transition region where strain changes quickly, the linewidth of X^0 is much larger (7.6 ± 0.3 meV) compared to the regions with linear strain gradient or no strain ($\sim 3.0 \pm 0.1$ meV).

We note that the characterization of strain presented in Fig. S11 and summarized in Fig. 2a, provides all the relevant details to understand dark exciton diffusion and it is consistent with the observed asymmetry in the exciton diffusion across the strained and unstrained regions. The precise characterization of strain over the full area would not add much to our analysis which is done on a line. Moreover, the size of our flake is quite large, approximately $12 \times 20 \mu\text{m}$, and a full characterization is challenging and we did not perform it initially. At this time, this particular sample has undergone several measurements and thermal cycles that have dramatically affected the original strain landscape. Therefore, performing this measurement now would not provide precise information.

Prompted by this comment, we modify Fig.2 and 3, we add the new Fig. S11 and a full discussion on strain in the SI.

Figure S11: **a** - PL map of the sample highlighting regions of uphill energy landscape (white dotted box) and flat energy landscape (yellow dotted box) generated by strain variation. **b** - Energy variation of X^0 measured along a horizontal line across the uphill and flat energy landscape. **c** - Red spectra are measured in the uphill energy landscape, blue spectra are taken across the interface of the strained and unstrained region, and black spectra are taken in the flat energy landscape. **d** - Spectra taken at very low excitation powers in different positions of the region of the sample indicated by the yellow dashed rectangular box. All spectra within the yellow dotted box show a small variation in energy X^0 of ~ 4 meV due to slight inhomogeneity, revealing a nearly flat energy landscape.

- The authors claim that the reason they detect the unshifted bright exciton, trion, and biexciton emission a few micrometers away from the excitation spot (Fig 4b, c) is due to their high NA and that they still pick up the emission from approx. 2.5 μm away (page 5 and 6). I assume there is a strain gradient even in the emission area of the diffusion spot (which has a size of about 2 microns). Do they see any line broadening due to the summation of different strain values? Can the authors estimate the strain variation in the diffusion spot of the bright excitons?

I wonder why they do not see broad line shapes for the dark excitons, which have a much longer diffusion length. In this case the large detection spot of more than 2.5 micrometer will cover emission areas of different strain values.

Here, the reviewer asks about possible line broadening observed in the exciton peaks due to the presence of large strain in our sample. Before answering in detail this question, we would like to stress that we used three different imaging methods for our measurements. These are described in detail in Note 1 and Fig.S1 of the SI.

Fig.2a is a PL map of the sample taken in a confocal mode using one galvanometer system and a spatial filter (lenses + pinhole) to select a small region of the sample. After the spatial filter, the exciton signal is routed to APDs. The spatial resolution achieved with this optical method is ~ 350 nm and allows for high quality PL maps of the sample. The polarization-resolved spectra reported in Fig.1, the power-dependent Fig.S7, and the space-dependent spectra of Fig.4a and Fig.S11 have been taken in confocal mode but instead of the APDs, the exciton emission is routed to the spectrometer. Therefore, the exciton spectrum is sampled over a spatial region of ~ 350 nm.

Fig. 2 c,d,e and Fig.3a,b,c are measurements of the exciton expansion and have been taken by directly imaging the far field emission with a EMCCD camera. The images produced by this method have lower resolution as discussed in the previous comments. No spectrum has been measured through this path.

In Fig.4b,c, the spectra have been taken by exciting the sample in one position with one galvanometer (GM1) system, and detecting the emission at a different position with the second galvanometer (GM2) system. The signal from the second galvanometer is routed through the same spatial filter and we expect the same high spatial resolution for this measurement. Therefore, in all the spectral measurements in most regions of the sample we do not expect significant line broadening due to the small sampling area that prevents the summation of emission from regions of the sample that are more than 350 nm apart. In regions of the sample where strain changes very quickly, such as the interface region between strained and unstrained regions (see Fig.S11 and previous comment), we do observe a broadening of the exciton peak.

In the transition region where strain changes quickly, the linewidth of X^0 is much larger (7.6 ± 0.3 meV) compared to the regions with linear strain gradient or no strain ($\sim 3.0 \pm 0.1$ meV).

A few considerations on the spectral resolutions of Fig.4b and c are in order. In Fig.4c we detect (with GM2) bright excitons emission coming from the excitation position which can be several μm away. In fact, the emission energy of bright excitons is constant as a function of the position. In this case, we expect some distortion of the detection that might reduce the spatial filtering

efficiency. Namely, when the detection system picks up light from a position different from the position of the galvanometer system, we might collect emission from excitons in areas larger than 350 nm. In particular, the further we collect from the excitation spot, the more distortion we expect. We check this effect, by comparing the FWHM of the emission from bright excitons measured when they are excited and detected in the same location (this is the case of Fig.4a where we excited with GM1 and detect with GM1), and the case where the bright exciton emission at the location of GM1 is picked up by the detection GM2 (this is the case of Fig.4c). In Figure R5 below we compared the FWHM as a function of the distance. While in Fig.4a the FWHM is basically constant, the FMHW in Fig.4c increases.

Figure R5: comparison of the FWHM of the bright exciton emission in Fig.4a and Fig.4c. In Fig. 4a emission and detection occur at the same location fixed by the position of GM1. In this case the resolution of the setup is fixed to ~350 nm. In Fig.4c, detection (GM2) occurs at increasing distance from the excitation (GM1). In this case the resolution on the signal picked by the GM2 changes as a function of distance and results in a larger FWHM.

We note that the detection of dark excitons in Fig.4c is different from the one of bright excitons. Emission of dark excitons is collected at the location of the galvanometer system and we expect an efficient spatial filtering. Therefore, there is no significant broadening in the emission linewidth of dark excitons.

Unfortunately we cannot provide an estimation of the strain from the broadening of the bright exciton peak. In Fig.4a, the sampling area of 350 nm is too small to provide any significant information. In Fig.4c, we do not know exactly the size of the sampling area. However, we would like to stress that the strain over the range of measurements of Fig.4 is fully characterized by extracting the strain value from the emission energy of the bright excitons as reported in the top panels of Fig.4.

- In figure 4 the authors excited at a low strain level and measured for increasing strain and the other way around. The authors should also excite at an intermediate strain level and measure spectra at lower and higher strain positions on the sample. In this case they can directly measure up and downhill diffusion in the same excitation scenario.

The Reviewer's description of the experiments shown In Figure 4 is correct. In Fig.4b, we measure the emission spectra by exciting at the bottom of the energy ramp (high strain) and detecting at increasing distance from the excitation. In Fig.4c, we measure the emission spectra by exciting at the top of the energy ramp (no strain) and detecting at increasing distance from the excitation. **We would like to stress that the experimental conditions are kept the same in the uphill and downhill measurements. Specifically, we use the same laser power and the measurements are performed in the very same region of the sample.** This approach allows us to study the diffusion of excitons in both uphill and downhill conditions, similarly to the ones suggested by the Reviewer. Our energy landscape had a length of 6 μm and a height of 20 meV, and our goal is to study exciton propagation over the maximum values of uphill and downhill conditions. For this reason, we choose to excite at the very top and very bottom of the energy ramp, which enables us to measure diffusion over longer distances and higher energy landscapes.

The Reviewer asks to perform the experiments by exciting at the center of the ramp (mid strain) and detect both uphill and downhill. However, for the reasons discussed above, we believe that this kind of experiment would not provide any further information as downhill and uphill diffusion are already extensively studied for in more comprehensive conditions. In the new version of the manuscript we add new data on exciton diffusion performed by varying the pump power and the energy landscape. These new measurements are now presented as Figure S15-17 in the SI. We believe that these new data provide a more comprehensive description of the dark exciton diffusion in different energy landscapes.

Furthermore, we have to note that the strain landscape is extremely delicate and it reduces after a few thermal cycles. For the initial measurements, we had to cool down our sample to 7K several times. The result is that the energy ramp has now changed due to release of the initial strain. Therefore, we cannot perform any additional measurements on this particular sample.

- In figure 4 the colored dots used to mark the excitons are hardly visible and make it difficult to match the spectral feature with the correct exciton species.

We thank the Reviewer for noting this issue with the visibility of the colored dots in Figure 4. The choice of small dots in Fig.4 was taken to not cover the experimental spectra. In particular, the emission from dark excitons at long distances is very weak. Moreover, the colors of the dots are chosen to separate the bright exciton species (red dots) with the different dark exciton complexes (yellow, blue, green, pink). To better highlight the position of the peaks and make it easier to match the spectral features with the correct exciton species, we now increase the size of the markerks in Fig.4.

- The authors explain their results due to the repulsion driven propagation of dark excitons caused by the strong interaction due to the permanent out-of-plane dipole (Page 4 Line 7). Can the authors explain this in more detail? For me it is not clear why the out-of-plane dipole of dark excitons should create a stronger exciton-exciton interaction as the in-plane dipole of the bright excitons. Can they give references? What is actually the assumed mechanism for the diffusion of the bright excitons if it is not exciton-exciton interaction? Nothing is mentioned in the manuscript.

We thank the Reviewer for this question that, together with similar questions from other Reviewers, has triggered significant improvements in the analysis of exciton-exciton interaction and its role in the diffusion properties. For a detailed discussion on this topic we refer the Reviewer to the **“General remarks”** at the beginning of our response. Particularly relevant for this question are: **“RECONSIDERATION OF DIPOLE-DIPOLE INTERACTION”** and **“NEW CALCULATIONS OF THE EXCITON DENSITY”**.

In summary, in the revised model, we identify exchange interaction as the predominant interaction mechanism and we neglect the dipole term. Exchange interaction U_{ex} is always positive and results in a total repulsive interaction energy $\Delta E = n_0 U_{ex}$ for excitons, where n_0 is the exciton density.

In our new calculations, we show that ΔE is much larger for dark excitons compared to bright excitons. This is confirmed by our experiments. The stronger repulsive interaction among dark excitons results in a longer diffusion length, defined as the distance that excitons can travel in their lifetime. However, diffusion can occur also with much weaker repulsive interaction. Diffusion is the transport of charges occurring due to non-uniform concentration. Clearly, strong repulsive interaction, as in the case of dark excitons, boosts this transport by providing additional kinetic energy.

A similar effect has been observed in interlayer excitons in heterostructures, including by Sun et al. Nature Photonics 16, 79 (2022) and Jauregui et al. Science 366, 875 (2019)) in which the repulsion due to dipole interaction results in longer diffusion.

To clarify these points in the manuscript, we add a comprehensive description of exchange interaction and the results of diffusion lengths as discussed in the general remarks at the beginning of this response. Moreover, when discussing the result on bright exciton diffusion in the manuscript we add the following sentence:

“The diffusion of bright excitons is driven by their non-uniform concentration at the excitation spot and it is constrained by the short lifetime, in the order of 1-5 ps, and weak interaction.”^{30,31} ”

- The authors write “...the diffusion of D- is expected to be limited by the smaller binding energy of the exciton carrier complex.” (page 6) This connection is not clear to me. Why does the diffusion depends on the binding energy? Can the authors give references.

In addition on page 5 they write “Despite its short lifetime X0 has a large binding energy and tensile strain generates a funneling effect...”. It is true that X0 has a large binding energy but how is it connected to the diffusion?

We thank the Reviewer for this question. However, we would like to note that the discussion of the experimental results of Fig.4 has been reformulated in light of the new interpretation based on exchange interaction, and the sentence highlighted by the Reviewer is no longer in the manuscript. The new interpretation is consistent with the observation of longer diffusion for longer binding energy. On one hand, the exchange interaction strength depends on the binding energy according to $U_{ex} \sim a_0^2 E_b$. This formula is derived in Ref. 42, 43, and 44 of the main text. Dark trions possess a smaller binding energy (~17meV) compared to dark excitons (~ 500meV), which means that the interaction potential strength is weaker for dark trions. On the other hand, weaker bound states will generally be more susceptible to disorder and impurities, which can limit their transport. This is because these weaker bound states are more likely to be disrupted by interactions with defects or other sources of disorder in the material, such as strain. The latter is the case of the experiments of Fig.4. Additionally, weaker bound states will be more sensitive to changes in the local dielectric environment, which can further limit their transport properties. Therefore, the binding energy is a relevant factor when studying transport of excitonic states in semiconductors.

In the new version of the manuscript, the sentence highlighted by the reviewer has been replaced by a completed description of the process of the excitation, transport and relaxation of dark excitons, that are now illustrated in the new Figure 5.

- Can the authors perform lifetime measurements to estimate the diffusion speed for the different exciton types?

We thank the Reviewer for the suggestion of performing lifetime measurements to estimate the diffusion speed of different exciton types in our study. In the general remark “MEASUREMENT OF THE DIFFUSION LENGTH” we discuss our results on the measurements of diffusion length and the calculation of diffusion constant which is an equivalent figure of merit to the diffusion speed. We estimate the diffusion constant of dark excitons from the measurements of the diffusion length according to $D = L_D^2 / \tau$, and using the lifetime values reported in the literature. For the purposes of estimation, we use a lifetime of 250 ps. We found that the diffusion constant ranges from 68 to 240 cm²/s in the unstrained direction (+x), while it ranges from 40 to 110 cm²/s in the strained uphill direction (-x). The diffusion length increases with excitation power caused by the repulsive interaction among excitons. Unfortunately, we do not have the experimental capabilities to conduct lifetime measurements. The temporal resolution of our setup is limited to ~350 ps which is not enough to capture the dynamics of excitons at low T in WS₂. However, we would like to note that Cadiz et al. (Appl. Phys. Lett. 112, 152106 (2018)) reported a diffusion constant for dark excitons in WSe₂ of ~250 cm²/s in agreement with our results.

We incorporate this information into the revised manuscript by adding a detailed discussion on the measurements of the diffusion length and the estimation of the diffusion constant.

- Page 6: what is meant by 'relax to their ground state' this state is not well defined in the text.

We thank the Reviewer for pointing out this unclear passage that we now improve. When we write that excitons relax to their ground state, we mean that they relax to the lowest energy state available to them at that particular position in the energy landscape. This ground state energy level is not necessarily well-defined but depends on the specific energy landscape of the system.

To clarify this, we add to the manuscript an illustration (Fig.5b) of the relaxation path of dark excitons after transport, and the following description: “The transport and relaxation pathways for the dark exciton D^0 are schematically illustrated in Fig. 5c. The system is initially excited to create dark excitons as shown in Fig.5a. The high density of dark excitons favor the strong exciton-exciton repulsive interaction that increases the overall energy of the dark exciton population at the excitation position. This repulsive interaction at the excitation position provides initial momentum to excitons which diffuse away even towards locations of the sample with higher energy landscape. The transported hot dark excitons can then thermalize toward the bottom of their energy dispersion via scattering with low energy Γ acoustic phonons.²⁸ From this state they can radiatively recombine in the form of a dark exciton D^0 , dark trions D^- , phonon replica, T_1 or I^0 .”

- What is the point of estimating the dark exciton density? This estimation appears rather unmotivated to me in the text. Is their result in line with their expectation, and how does it differ for the bright exciton density?

We thank the Reviewer for this question that has triggered further theoretical calculations that have improved the overall quality of our work.

In our study, we estimate the density of dark excitons because the effective interaction energy between excitons depends on their density according to $\Delta E = n_0 U_{ex}$, where n_0 is the exciton density and U_{ex} is the interaction strength between two excitons. Exciton density is a crucial factor in understanding the interactions between excitons in the system. We experimentally estimated the lower limit of dark exciton density in our manuscript. However, to further understand exciton interactions and provide additional insights, we include in the revised manuscript a simple theoretical model of coupled rate equations to estimate the density of bright and dark excitons. This model considers both radiative and non-radiative decay pathways and allows us to estimate the density of dark excitons in our system. **We find good agreement between the measured and calculated dark exciton density.**

The model returns that the density of dark excitons is orders of magnitude higher than the one of bright excitons, suggesting a much stronger total interaction for dark excitons due to their high density. This result aligns with our expectation that the strong exciton-exciton repulsion observed in our experiment is primarily due to exchange interactions between dark excitons. In summary, the estimation of the dark exciton density is crucial for understanding the interaction of dark excitons and overall dynamics of the system.

For a full discussion of the new theoretical models, the comparison with the experimental results, and the list of changes to the manuscript we refer the Reviewer to the general remarks "NEW CALCULATIONS OF THE EXCITON DENSITY".

- The authors claim that they used a new fabrication technique in order to produce a sample with a strain gradient. This is not true. In contrast such strain landscapes appear often naturally in transferred layers since a contact angle is often unavoidable. Samples with large strain gradients were already used before (Nature Communications 12, 7221 (2021))

We thank the Reviewer for bringing up this point and reference to our attention. We apologize for any confusion and would like to clarify that, while our fabrication technique may not be entirely new, it is a variation that allows us to achieve a specific strain gradient profile that is necessary for our study. In particular, our technique produces an almost perfectly linear strain gradient over a 6 μm distance with a 20meV energy landscape height, which is different from the strain gradients observed in transferred layers due to pillar effects such as in Nature Communications 12, 7221 (2021). In the latter work, the strain distribution extends over a comparable distance but it is very inhomogeneous and the study on exciton funneling is performed only over 2 μm .

Prompted by this comment, we tone down the description of our fabrication technique. We change the sentence to: "Due to the introduction of an advanced fabrication process and imaging setup, we are able to study diffusion of dark excitons in a linear potential energy gradient." We also add the paper Nature Communications 12, 7221 (2021) as Ref. 18.

- In their introduction the authors mention dynamics of interlayer excitons. However, citations regarding the (strain) dependent excitons diffusion of momentum dark excitons are missing. E.g. Nature Communications 12, 7221 (2021).

We thank the Reviewer for pointing out this relevant citation that was missing from the initial version of our paper. We change the sentence in the introduction and include this reference (Ref. 18) to our manuscript: "Although dynamics and transport of interlayer excitons in van der Waals heterostructures¹⁵⁻¹⁷ and intervalley excitons¹⁸ have been recently investigated, spin-forbidden dark excitons dynamics and their transport properties have not been fully explored yet,¹⁹ due to the challenges associated with their detection using far-field spectroscopy techniques.^{11,20,21}"

- The schematics figure 1a can lead to misunderstandings. Why are the dark excitons depicted above the monolayer floating in the air?

We thank the Reviewer for this feedback on Figure 1a. This cartoon illustrates the schematic of our experiments in which dark excitons can propagate over long distance and uphill energy landscapes due to their strong interaction energy. The “floating in the air” of dark excitons indicates that they have more energy compared to bright excitons which are stuck to the excitation position. To better indicate that dark excitons acquire energy due to repulsive interactions, we add a more visible energy axis to the cartoon along the z direction, similarly to Figure 1 of Nature Communications 12, 7221 (2021). It is important to us that the figures in our paper accurately represent the concepts discussed in the text to avoid any potential misunderstandings or confusion for the readers.

Reviewers' Comments:

Reviewer #1:

Remarks to the Author:

The efforts of Chand and co-authors have resulted in a revised manuscript that improved since the initial submission. There are still a few comments that need to be addressed, however, before I can fully recommend the paper for publication in Nature Communications.

1) Regarding the long diffusion length of the intralayer bright excitons, I appreciate the authors' efforts in describing the steady state observations using the diffusion length. This analysis is useful as it provides a quantitative comparison of the bright and dark state diffusion. As the authors correctly point out, the work by Cadiz et al. also extracted a diffusion length of 1.5 μm for bright excitons, but in their data the short decay (0.2 μm) plays a very important role. If you compare Figure 2e of your manuscript with figure 2c of Cadiz et al, APL (2018), there is a clear difference between the two. This is what I was referring to in the previous report: The bright exciton cloud should closely match the laser excitation profile given the short lifetimes at low temperatures.

Could this be due to the way that the exciton cloud is measured – bypassing the confocal microscope and using the EMCCD? As the authors mention in the response to Reviewer #3, this imaging modality might make sharp edges more diffuse, explaining why you observe some emission outside of the area of the flake. Looking at figure S13b, which I assume was taken with the galvo-mirror-2 (GM2) confocal path, it seems like the decay of PL intensity of the bright exciton is much faster (with increasing distance), which would be consistent with the fast lifetime of the bright excitons. Can the authors collect spatial data on the bright exciton emission by fixing the laser spot with GM1, and sweeping the collection path with GM2 while reading out (filtered) APD counts? Does that show a smaller intralayer exciton cloud?

The authors should address this issue in the main text, and/or provide other potential explanations.

Also, the sentence added to the manuscript is slightly misleading and should be modified: "This surprisingly large value for the diffusion length of bright excitons has been already reported".

2) The exchange interaction strength for the dark and bright excitons that the authors estimate seems large by over an order of magnitude when compared to previous experimental works: Barachati et al. Nature Nanotechnology 13, 906–909 (2018) for WS₂ excitons (extracted from exciton polariton measurements), and Scuri et al. PRL 120, 037402 (2018) in MoSe₂. Potential discrepancies with theory are also discussed briefly in the Barachati paper. The authors should comment in the main text or the SI, providing a range of values for their estimate of the interaction energy and dark exciton density.

Also, in the estimation the exchange energy, the authors should use experimental data for the Bohr radius and binding energy, where possible. See for example: Chernikov et al, PRL 113, 076802 (2014) and Stier et al Nature Communications 7, 10643 (2016).

Reviewer #2:

Remarks to the Author:

The authors have strongly modified the manuscript, in particular reconsidering the role of the dipole-dipole interaction for the observed density-dependent (uphill) diffusion of the dark exciton, which is now attributed to the exchange interaction. The changes have improved the analysis of the results. However some questions on the new interpretation need to be answered before publication in Nature Communications.

1- While the authors write that the different density-dependent behaviour of peaks X0 and D0 is "attributed to a higher density" of the spin-dark excitons compared to the bright ones, more comments should be dedicated to peak I0. This peak is emitted from momentum-dark excitons, with energies roughly 30 meV below the energy on the bright states, while being only few meV

above the energy of the spin-dark excitons (6.5 meV in these experiments). As a consequence, one would expect the momentum-dark excitons to have a density much higher than the bright ones, while relatively closer to the one of spin-dark states. The authors should comment if this relatively high density of momentum-dark excitons could be the case in their experiment and if it could have an impact on transport (also in relation to Figs. 5(b) and 4c, respectively).

2- While the density-dependent transport is now explained via the exchange interaction, more comments should be added on other mechanisms, e.g. the Coulomb-hole contribution to the renormalization (potentially attractive in monolayers, see Ref. [43]) or the Auger recombination, potentially important for transport at high exciton densities.

Reviewer #3:

Remarks to the Author:

I thank the authors for addressing all of my concerns in great detail and for including all comments as well from the other referees.

The new theoretical model considering the exchange interaction instead of the dipole-dipole interaction in combination with the exciton density calculation is convincing. Together with the additional measurements on the strain profile and the diffusion length the paper has significantly improved.

- I appreciate that the authors moved the power dependent spectra to the SI
- Together with the updated Fig.2 showing the measurements areas and the detailed explanation on different measurement techniques as well as the new Figure S11 my concerns (linewidth, strain landscape, diffusion...) are properly addressed.
- I understand that the sample undergoes changes with time (strain relaxation) and that additional measurements are not possible. With the current changes in the manuscript, I agree with the authors that additional measurements are not required.

Therefore, I can now recommend publication.

REVIEWER COMMENTS

Reviewer #1 (Remarks to the Author):

The efforts of Chand and co-authors have resulted in a revised manuscript that improved since the initial submission. There are still a few comments that need to be addressed, however, before I can fully recommend the paper for publication in Nature Communications.

1) Regarding the long diffusion length of the intralayer bright excitons, I appreciate the authors' efforts in describing the steady state observations using the diffusion length. This analysis is useful as it provides a quantitative comparison of the bright and dark state diffusion. As the authors correctly point out, the work by Cadiz et al. also extracted a diffusion length of 1.5 μm for bright excitons, but in their data the short decay (0.2 μm) plays a very important role. If you compare Figure 2e of your manuscript with figure 2c of Cadiz et al, APL (2018), there is a clear difference between the two. This is what I was referring to in the previous report: The bright exciton cloud should closely match the laser excitation profile given the short lifetimes at low temperatures.

Could this be due to the way that the exciton cloud is measured – bypassing the confocal microscope and using the EMCCD? As the authors mention in the response to Reviewer #3, this imaging modality might make sharp edges more diffuse, explaining why you observe some emission outside of the area of the flake. Looking at figure S13b, which I assume was taken with the galvo-mirror-2 (GM2) confocal path, it seems like the decay of PL intensity of the bright exciton is much faster (with increasing distance), which would be consistent with the fast lifetime of the bright excitons. Can the authors collect spatial data on the bright exciton emission by fixing the laser spot with GM1, and sweeping the collection path with GM2 while reading out (filtered) APD counts? Does that show a smaller intralayer exciton cloud?

The authors should address this issue in the main text, and/or provide other potential explanations.

We are grateful to the Reviewer for carefully reading our manuscript, for appreciating our previous modifications, and for providing further helpful feedback. We agree that the comparison with Cadiz et al. (APL 2018) is useful, and we thank the reviewer for pointing out the differences in the observed bright exciton clouds between the two studies. To investigate the possibility that the measured exciton cloud could be affected by a particular experimental technique, we follow the Reviewer's suggestion to look at the spatial distribution of the bright exciton emission when collected in the confocal mode, namely by exciting the system through GM1 and collecting exciton emission by scanning around with GM2.

We have carried out this experiment and found that the results are indeed more consistent with Cadiz et al., as predicted by the Reviewer. Figure R1 shows the profile of the laser and of the bright exciton cloud measured with APDs in confocal mode using two decoupled galvanometer

systems. The normalized intensity of the exciton overlaps with the laser profile down to 10^{-2} PL counts (see Figure R1c), similarly to what observed by Cadiz et al. Moreover, the estimation of the diffusion length away from the excitation spot in this measurement returns a smaller value compared to the EMCCD with $L_D = 0.8 \mu m$. These observations suggest two main conclusions.

On one hand, the measurements with the EMCCD are more sensitive and can detect weaker signals, such as the one of the slow decay of bright excitons. This signal is not fully detected by the APDs used in the confocal mode that, due to limited sensitivity, mostly pick up the signal from the faster and stronger decay. Note, however, that a small tail of slow decay is still present. On the other hand, the measurement of the exciton clouds with the EMCCD could be affected by errors intrinsic to the imaging modality that could make point sources more diffuse. In conclusion, our estimate of the diffusion length of bright excitons, when measured with the EMCCD, includes both fast and slow decay but provides a lower resolution compared to confocal methods. As a final note, we would like to stress that we have initially performed diffusion measurements in confocal modes, but due to the limited sensitivity we have eventually preferred far field imaging with the EMCCD that provided better results in the detection of dark excitons.

We update the manuscript to include this discussion. Specifically, we add to section 5 of the SI the results of the experiment suggested by the Reviewer with the new Figure S11. In the same section we discuss the potential effects of the imaging modality on the observed exciton clouds. The new text reads:

“Measurements of exciton diffusion are also carried out in confocal mode in which the spatial distribution of the exciton emission is collected with APDs by exciting the system through GM1 and scanning around with GM2 to record the PL emission. Differently from the measurements performed by recording the far field emission with the EMCCD (Fig. 2, 3, S8, S9), the confocal mode allows for better spatial resolution but lower sensitivity. Figure S11 compares the emission of the laser and the bright exciton cloud measured in confocal mode. The normalized intensity of the bright exciton overlaps with the laser profile down to 10^{-2} PL counts (see Figure S11c) and returns a diffusion length for bright excitons of $L_D = 0.8 \mu m$. There are some differences between the two diffusion measurements of X^0 obtained with the different imaging methods. While the EMCCD measurements have lower spatial resolution, they are highly sensitive and can detect even weak signals, such as the slow and weak decay of bright excitons, or the dark exciton emission. The APDs used in the confocal mode result in higher spatial resolution but have limited sensitivity and can mostly capture the faster and stronger decay signal of X^0 .”

Figure R1: Diffusion measurements with confocal imaging at $T = 7\text{K}$. **a** - Spatial profile of the laser imaged with APDs by exciting through GM1 and scanning around with GM2 to record the PL emission. **b** - Bright exciton diffusion measured in confocal mode by exciting the system through GM1 and scanning around with GM2 to record the PL emission. **c** - Comparison of the normalized intensity profile of the laser and the bright excitons emission extracted from **b** and **c**.

Also, the sentence added to the manuscript is slightly misleading and should be modified: “This surprisingly large value for the diffusion length of bright excitons has been already reported”.

Following the Reviewer’s suggestion, we modify this sentence to better reflect the comparison with Cadiz et al. and include the effects due to the imaging techniques discussed above.

“A similar value for the diffusion length of bright excitons has been already reported¹⁹ and attributed to the combined contribution of a rapid and slower decay rate.³⁶ However, we note that, while our imaging method is capable of detecting very weak signals, point sources in the sample can appear diffuse, affecting the spatial resolution. A comparison with confocal imaging is discussed in Section 5 and Figure S11 of Supplementary Information.”

2) The exchange interaction strength for the dark and bright excitons that the authors estimate seems large by over an order of magnitude when compared to previous experimental works: Barachati et al. Nature Nanotechnology 13, 906–909 (2018) for WS2 excitons (extracted from exciton polariton measurements), and Scuri et al. PRL 120, 037402 (2018) in MoSe2. Potential discrepancies with theory are also discussed briefly in the Barachati paper. The authors should comment in the main text or the SI, providing a range of values for their estimate of the interaction energy and dark exciton density.

Also, in the estimation the exchange energy, the authors should use experimental data for the Bohr radius and binding energy, where possible. See for example: Chernikov et al, PRL 113, 076802 (2014) and Stier et al Nature Communications 7, 10643 (2016).

We thank the Reviewer for this valuable comment that has triggered further analysis of our results. The estimation of the Reviewer of the interaction strength is correct. Barachati et al. reported an interaction strength for the bright excitons of $6 \cdot 10^{-13} \text{ meV cm}^2$, while Scuri et al of $9.6 \cdot 10^{-13} \text{ meV cm}^2$. With our data, we can extract the interaction strength U_{ex} of dark excitons from $\Delta E = n_0 U_{ex}$ assuming a blueshift $\Delta E = 20 \text{ meV}$ and using the dark exciton density n_0 calculated with our model of $8 \cdot 10^{11} \text{ cm}^{-2}$. For dark excitons, we obtain $U_{ex} \sim 2.5 \cdot 10^{-11} \text{ meV cm}^2$, which is about one order of magnitude larger than the values reported previously for bright excitons. Please note that, in the estimation of the generation rate, we update the laser spot size on the sample following the new measurements in confocal mode suggested by the Reviewer in the previous comment. We use the central part of the laser spot which is more relevant in the excitation generation at high pump powers.

When comparing the experimental value of the interaction strength with the theoretical prediction $U_{ex} \sim a_o^2 E_b$, we note that different values of the exciton radius and the binding energy are reported in the literature from both experimental measurements and theoretical calculations. The table below summarizes some of the experimental ones for bright excitons.

	Experiments
E_b (eV)	0.32 (PRL 113, 076802 (2014)), 0.41 (Nat Commun 7, 10643 (2016)) 0.70 (Nature 513, 214–218 (2014))
a_o (nm)	1.53 (Nat Commun 7, 10643 (2016))

We also note that the analytical formula for the interaction strength includes a prefactor that depends on the shape of the exciton wavefunction as predicted in PRB 96, 115409 (2017), and discussed in Barachati et al. Therefore this approach is only valid to estimate the order of magnitude. However, by neglecting the prefactor (as previously done by Scuri et al.) we can define a range of values for the interaction strength: $7.5 \cdot 10^{-12} - 1.6 \cdot 10^{-11} \text{ meV cm}^2$. The order of magnitude of the interaction strength estimated from our experiments is comparable to the theoretical ones.

Furthermore, by using these values for the interaction strength, we can estimate a range of experimental values for the dark exciton density of $1 \cdot 10^{12} - 3 \cdot 10^{12} \text{ cm}^{-2}$. We note that this range is just slightly higher than the value of the dark exciton density of $8 \cdot 10^{11} \text{ cm}^{-2}$ calculated with our model, but it is overall comparable.

In conclusion, our values of the interaction strength of the dark excitons are larger than the ones measured for bright excitons, but generally comparable with the theoretical estimation. We can identify a few reasons for the discrepancies in these values. Here we are comparing dark and

bright excitons that have different size, binding energy and potentially wavefunction shape. These differences, although small, can result in different interactions. To the best of our knowledge, there are no experimental values for the dark exciton radius, thus a direct comparison with the analytical values is impossible at this time. Moreover, our estimate of the interaction strength depends on the calculation of the dark exciton density that comes from our theoretical model, that, as discussed in the SI, is approximated and only takes into account the main relaxation processes. Although our model returns compatible values for the exciton density, small errors could affect the estimate of the interaction strength.

To account for this new discussion, we update the discussion on the estimation of the dark exciton density and add the value for the interaction strength. We add as references the papers indicated by the Reviewer. We add the following text to the main text:

“For a blueshift of $\Delta E = 20 \text{ meV}$ as in the case of Figure 4b,c, the corresponding interaction strength is $U_{ex} \sim 2.5 \cdot 10^{-11} \text{ meV cm}^2$, which is one order of magnitude larger compared to the one measured in bright excitons.⁵⁰ This deviation can originate from the different properties between dark and bright excitons, or from the limitations of the model used to estimate the dark exciton density discussed in the SI. However, the calculated density is compatible with the ones extracted from the experimental data when the interaction strength U_{ex} is estimated by using the values of the bright exciton radius and binding energy reported in the literature (more details in the SI).^{45,51,52} The estimated range of experimental values for the dark exciton density is $1 \cdot 10^{12} - 3 \cdot 10^{12} \text{ cm}^{-2}$.”

We add to Note 7 of the SI the following sentence:

“A range of values for the interaction strength of bright excitons of $7.5 \cdot 10^{-12} - 1.6 \cdot 10^{-11} \text{ meV cm}^2$ can be estimated from the reported experimental values of the exciton binding energy (ranging from 0.32 to 0.7 eV) and the measured Bohr radius of 1.53 nm.^{23,25,26}”

We thank the Reviewer again for the valuable input, which has helped us to improve the quality of our manuscript. We hope the Reviewer will find our answer satisfactory and can now fully recommend publication of our work.

Reviewer #2 (Remarks to the Author):

The authors have strongly modified the manuscript, in particular reconsidering the role of the dipole-dipole interaction for the observed density-dependent (uphill) diffusion of the dark exciton, which is now attributed to the exchange interaction. The changes have improved the analysis of the results. However some questions on the new interpretation need to be answered before publication in Nature Communications.

1- While the authors write that the different density-dependent behaviour of peaks X^0 and D^0 is "attributed to a higher density" of the spin-dark excitons compared to the bright ones, more comments should be dedicated to peak I^0 . This peak is emitted from momentum-dark excitons, with energies roughly 30 meV below the energy on the bright states, while being only few meV above the energy of the spin-dark excitons (6.5 meV in these experiments). As a consequence, one would expect the momentum-dark excitons to have a density much higher than the bright ones, while relatively closer to the one of spin-dark states. The authors should comment if this relatively high density of momentum-dark excitons could be the case in their experiment and if it could have an impact on transport (also in relation to Figs. 5(b) and 4c, respectively).

We thank the Reviewer for appreciating our efforts in addressing her/his previous comments and for the time and effort spent in reviewing our manuscript.

We also thank the reviewer for this comment on the potential impact of momentum-dark excitons on transport and their density. While it is intuitive to assume that momentum-dark excitons (I^0) may have a high density due to their energy being close to that of spin-dark excitons (D^0) and lower than bright excitons, our theoretical model shows that the density of I^0 is comparable to that of bright excitons (X^0) despite their lower energy and longer lifetime. The density of I^0 is shown with the blue curve in Fig. 5b and Fig. S15. To better highlight the value of the density of I^0 we add a zoom of the plot as inset in Fig.S15. The lower density of I^0 compared to D^0 can be explained by considering the timescale of the formation and decay channels available to I^0 . The formation of I^0 occurs through the $X^0 \rightarrow I^0$ channel mediated by scattering with K_3 phonons with rate $R_{K_3} = 2.5 \text{ ps}^{-1}$. The depopulation of I^0 occurs through the $I^0 \rightarrow D^0$ scattering channel with a slightly faster rate $R_{ID} = 2.75 \text{ ps}^{-1}$. Moreover, due to the long lifetime of I^0 (200 ps), the $I^0 \rightarrow D^0$ process dominates the population dynamics and its density is mostly limited by this faster decay channel towards D^0 . On the contrary, D^0 is the lowest exciton state with no other decay channels than the long radiative lifetime, and works as a relaxation bucket for all other excitonic species and high density of dark excitons can accumulate. Therefore, the interaction and transport of I^0 is expected to be smaller compared to D^0 . It is worth noting that the scattering channel $D^0 \rightarrow I^0$ is predicted to open only when the exciton kinetic energy increases (theory in Ref. 47). This is the reason why I^0 is observed propagating despite the low density. This is discussed in the main text at the beginning of pag.8.

To better clarify the point highlighted by the Reviewer, we modify the sentence at the beginning of pag. 8 to include this discussion. It reads:

“It is worth noting that despite the similar lifetime of I^0 and D^0 , our model returns a density for I^0 which is orders of magnitude smaller than the one of D^0 . While D^0 is the lowest exciton state of the system, the dynamics of I^0 is governed by the fast $I^0 \rightarrow D^0$ scattering channel that quickly depopulates I^0 in favor of D^0 . Interestingly, in the experiment of Fig.4c, I^0 recombines at higher energy with respect to the excitation location despite its low density and weak interaction. Even though it is energetically unfavorable, the transition $D^0 \rightarrow I^0$ can occur when the kinetic energy of dark excitons is larger than ~ 20 meV,⁴⁷ compatible to the one observed in our experiments. This observation suggests that I^0 generates from the relaxation of the transported D^0 .”

2- While the density-dependent transport is now explained via the exchange interaction, more comments should be added on other mechanisms, e.g. the Coulomb-hole contribution to the renormalization (potentially attractive in monolayers, see Ref. [43]) or the Auger recombination, potentially important for transport at high exciton densities.

We thank the Reviewer for this comment, and we agree with her/him that these mechanisms can be potentially important for transport and should be included in the discussion. In the theoretical study mentioned by the Reviewer (Ref. 43), the direct interactions between spin-allowed $KK-K\Lambda$ and $KK-KK'$ excitons are predicted to be attractive, while the direct interaction between $KK-KK$ excitons is still repulsive. We can assume a similar dynamics for spin-forbidden excitons. However, the strength of these direct interactions is much weaker than exchange interactions. Therefore, in a first approximation, their role in the interaction-driven transport can be neglected. Nevertheless, at finite momentum, the strength of direct interaction between KK and $K\Lambda$ increases significantly. However, in WS_2 , the formation of $K\Lambda$ excitons requires further mechanisms, such as compressive strain (demonstrated experimentally in Ref. 38), thus their density in the sample is expected to be small.

The impact of Auger recombination is considered by including exciton-exciton annihilation in the theoretical model used to calculate the exciton density. The results in Figure 5b show that, at high generation rates ($> 10^{11} \text{ cm}^{-2} \text{ ps}^{-1}$), the dark exciton density, and thus transport, is significantly affected by Auger recombination. However, in the range of generation rates used in our experiments ($\sim 10^9 \text{ cm}^{-2} \text{ ps}^{-1}$) our calculations return a linear behavior suggesting that the effect of Auger recombination is weak.

To include this discussion we modify the main text as follows.

- We change the sentence discussing the role of direct interaction with: “While the direct contribution to the interaction is generally weak and vanishes for small values of center of mass momentum, the exchange contribution is always repulsive and finite,^{43,44} with an

approximate strength of $U_{ex} \sim a_o^2 E_b$, where E_b is the binding energy and a_o is the Bohr radius.”

- When discussing the results of the density model in Fig. 5b, we add: “We note that the effect of exciton-exciton annihilation is only significant for values of generation rates much larger than the ones used in the experiments.”

We add a more detailed discussion on the possible contribution of Coulomb attractive interaction and Auger recombination to the density-dependent transport in Section 6 of the SI: “Other interaction mechanisms other than exchange interaction could potentially impact transport, including direct interaction and exciton-exciton annihilation. The direct interactions between spin-allowed $KK-KA$ and $KK-KK'$ excitons have been predicted to be attractive, while the direct interaction between $KK-KK$ excitons is repulsive.²¹ We can assume a similar dynamics for spin-forbidden excitons. However, the strength of these direct interactions is much weaker than exchange interactions.²¹ Therefore, in a first approximation, their role in the interaction-driven transport can be neglected. The impact of Auger recombination is considered by including exciton-exciton annihilation in the theoretical model used to calculate the exciton density. The results in Figure 5b and S15b show that, at high generation rates ($> 10^{11} \text{ cm}^{-2} \text{ ps}^{-1}$), the dark exciton density, and thus transport, is significantly affected by Auger recombination. However, in the range of generation rates used in our experiments ($\sim 10^9 \text{ cm}^{-2} \text{ ps}^{-1}$), our calculations return a linear behavior suggesting that the effect of Auger recombination is weak.”

To better highlight the value of the density of I^0 , we add as an inset in Fig.S15b a zoom of the curves around the generation rate used in the experiments.

Reviewer #3 (Remarks to the Author):

I thank the authors for addressing all of my concerns in great detail and for including all comments as well from the other referees.

The new theoretical model considering the exchange interaction instead of the dipole-dipole interaction in combination with the exciton density calculation is convincing. Together with the additional measurements on the strain profile and the diffusion length the paper has significantly improved.

- I appreciate that the authors moved the power dependent spectra to the SI
- Together with the updated Fig.2 showing the measurements areas and the detailed explanation on different measurement techniques as well as the new Figure S11 my concerns (linewidth, strain landscape, diffusion...) are properly addressed.
- I understand that the sample undergoes changes with time (strain relaxation) and that additional measurements are not possible. With the current changes in the manuscript, I agree with the authors that additional measurements are not required.

Therefore, I can now recommend publication.

We thank the Reviewer for appreciating our efforts in addressing her/his previous comments and for recommending publication of our work. We really appreciate the thorough and fruitful review and feedback, which helped us to improve our work. We thank the Reviewer again for the time and efforts spent in reviewing our manuscript.

Reviewers' Comments:

Reviewer #1:

Remarks to the Author:

The authors responded well and thoroughly to all the comments. I therefore recommend the paper for publication in Nature Communications.

Reviewer #2:

Remarks to the Author:

The authors have carefully considered my report and added accordingly new comments in the paper. The manuscript has further improved and I can hence recommend publication.